# Flow chamber staining modality for real-time inspection of dynamic phenotypes in multiple histological stains

**Zhongmin Li** [ORCID]*, **Goetz Muench, Silvia Goebel, Kerstin Uhland, Clara Wenhart, Andreas Reimann**

Advancecor GmbH, Martinsried, Germany

* li@advancecor.com

**Data Availability Statement:** All relevant data are within the paper and its Supporting Information files.

## Abstract

Traditional histological stains, such as hematoxylin-eosin (HE), special stains, and immuno-fluorescence (IF), have defined myriads of cellular phenotypes and tissue structures in a separate stained section. However, the precise connection of information conveyed by the various stains in the same section, which may be important for diagnosis, is absent. Here, we present a new staining modality—Flow chamber stain, which complies with the current staining workflow but possesses newly additional features non-seen in conventional stains, allowing for (1) quickly switching staining modes between destain and restain for multiplex staining in one single section from routinely histological preparation, (2) real-time inspecting and digitally capturing each specific stained phenotype, and (3) efficiently synthesizing graphs containing the tissue multiple-stained components at site-specific regions. Comparisons of its stains with those by the conventional staining fashions using the microscopic images of mouse tissues (lung, heart, liver, kidney, esophagus, and brain), involving stains of HE, Periodic acid–Schiff, Sirius red, and IF for Human IgG, and mouse CD45, hemoglobin, and CD31, showed no major discordance. Repetitive experiments testing on targeted areas of stained sections confirmed the method is reliable with accuracy and high reproducibility. Using the technique, the targets of IF were easily localized and seen structurally in HE- or special-stained sections, and the unknown or suspected components or structures in HE-stained sections were further determined in histological special stains or IF. By the technique, staining processing was videoed and made a backup for off-site pathologists, which facilitates tele-consultation or -education in current digital pathology. Mistakes, which might occur during the staining process, can be immediately found and amended accordingly. With the technique, a single section can provide much more information than the traditional stained counterpart. The staining mode bears great potential to become a common supplementary tool for traditional histopathology.

## 1. Introduction

Immunofluorescence (IF) is a widely used immunohistological technique that renders the tissue constituents or the epitopes of cells of interest visible for microscopic analysis. But the

**Funding:** The author(s) received no specific funding for this work.

**Competing interests:** The authors have declared that no competing interests exist.

localization or knowledge of structures of the targets, which could assist in determining disease progression and directions for appropriate treatments in clinic and biomedical research, is challenging with traditional histological processing. In addition, hematoxylin and eosin stain (HE), one of the most general tissue staining methods routinely offers the status of cells and tissues in biological research and clinic for disease diagnosis. But unknown or suspected components or structures displayed in the HE stained section, which are often encountered by analysts and might deter making an accurate diagnosis for suspected cases, are also difficult to be defined currently.

To improve the current restrictions, HE or histological special stains are performed in the adjacent IF-stained section of the sequential. The location and structures can be determined with the collocation of the targets by merging images of both stains. However, serial sectioning may cut through the region of interest or may result in the loss of the regions necessary for critical diagnosis. This is particularly an issue with an inspection of high magnification, with a smaller region of interest that is of limited size and number, or in such a case as the tissue samples have been depleted or not enough material is available in stored blocks for serial sections.

Considering some of these limitations, computational staining techniques known as virtual staining have been developed recently. Using deep learning, virtual staining has been applied to label-free histological sections using various modalities such as autofluorescence [1–3], transformation framework [4], hyperspectral imaging [5], quantitative phase imaging [1], and multi-photon imaging [6]. The recent emergence of multiplexed tissue imaging techniques [7], such as CODEX, has also enabled researchers to overcome these limitations. The virtual staining of label-free tissue or the multiplexed tissue imaging techniques not only can reduce costs and allow for faster staining but also allows the user to localize the signals by performing further multiple manual or virtual stains on a single tissue section, where the destructive additional sectioning and staining process is avoided [8].

However, as most pathologists are mainly trained to perform diagnoses using a standard staining workflow and histologically stained specimens, the virtual images generated using alternative staining mechanisms (e.g., machine learning algorithms) or the new multiplexed imaging techniques might require additional training for analysts. Besides access to the digital staining matrix with deep learning or the expensive multiplexed staining platforms is not available in every laboratory and thus a method, which has widespread applicability, could be much appreciated currently.

For a widely practical approach that can be used without the complicated machine-deep-learning technique and in conformation to traditional tissue staining workflow, the procedures of bleaching and re-staining technique have been reported [9–12]. Multiplexed imaging has been also demonstrated by eluting or stripping antibodies with low pH or denaturation [13–15]. The major advantage of the approaches is that they would alleviate the need for additional sequentially sectioned slides. But an obvious shortage with the method is the iterated procedures spent on decoverslipping for rehydration and staining, and dehydration for coversliping and imaging, which might enhance the risk of section dropping. In addition, the biggest challenge is the difficulty in blending images of the multiple stains, which is important in aid of determining accurate diagnosis and disease progression.

A flow chamber setup, which is originally designed to create wall shear stress imposed by liquid flow in vitro and to real-time observe site-specific morphological changes of cells coated on coverslips, can become an efficient staining platform, in which the liquid is replaced with different staining solutions, and histological sections are substituted for the coating objects. The replacements would turn the setup into a flow chamber staining (FCS) device (outlined in Fig 1), allowing researchers to capture different detailed manifestations of morphology and components at site-specific regions, while the different stains contact with the histological

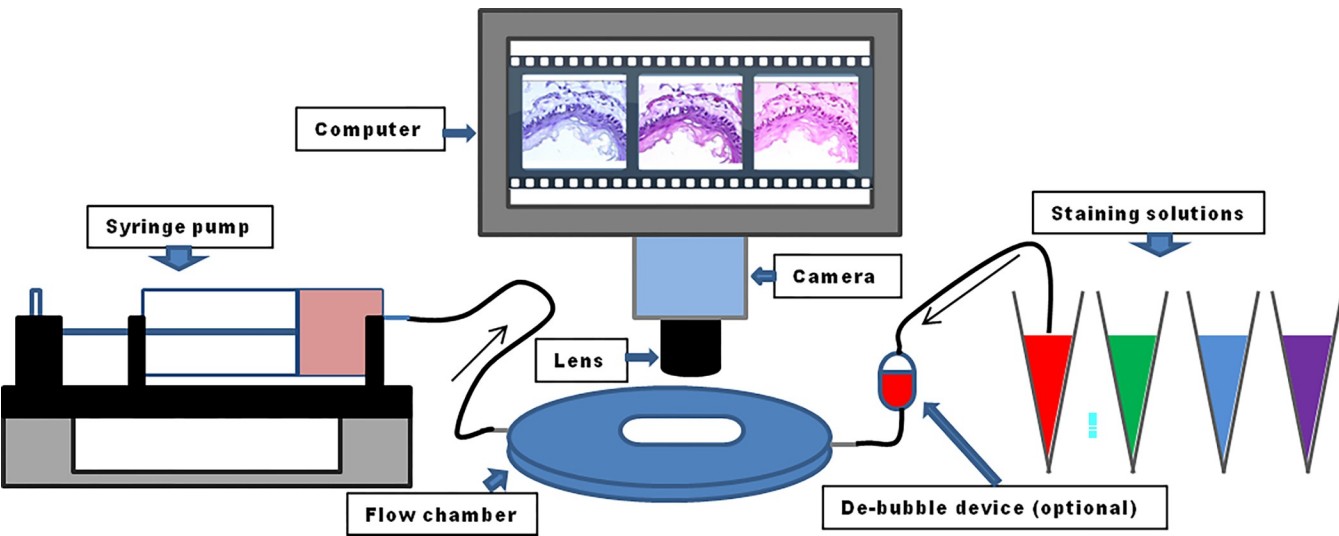

**Fig 1. Schematic diagram of parallel-plate flow chamber staining system.** The flow chamber system mainly consists of a syringe pump, tubings, a flow chamber, and fluid reservoirs of stain. The pump withdraws the liquid from the reservoir into the chamber via the tubings. Real-time inspecting and recording images of the events during perfusion are performed through an imaging system of microscope, camera, and computer.

sections. With FCS, the kinetics of discoloration and re-stained phenotypes at the site-specific regions of interest is real-time inspected and digitally captured under the microscope in each round of stain. More importantly, the orientation of the targeted region would be ensured for blending the multiple stains. The signal structures of IF stains could be easily disclosed and localized with FCS after restaining the IF-stained sections with histological stains.

To test the practical value of FCS, we performed histological stains by FCS, namely IF, HE and Sirius red in one single section (Fig 2A). The combination of the images acquired displayed a delicate distribution of various quantified target molecules or cells shown in IF, in addition to the general morphological structures seen in HE and the collagen reflected in Sirius red stain. We also carried out HE and then Periodic acid Schiff (PAS) for polysaccharides, mucosubstances (e.g., glycoproteins, glycolipids) and mucins, and Sirius red for collagen in a section (Fig 2B), which mimics the situation of diagnostic workflow in laboratories. For the same reason, by FCS HE stain was first made and the suspected components or structures displayed in the HE-stained section were then detected with IF (Fig 2C). At the end of the experiments, the HE-stained section can be archived by mounting medium on a slide for future possible investigations.

## 2. Results

### Validation of flow chamber setup

It is a prerequisite to assure there are no shear impairments imposed by chamber flow on the integrity of histological sections for the application of FCS. To check the impact of flow on the structure of sections, we performed 2-hour perfusion, using a plotted flow velocity of 1 ml/min, which would be used for the following experiments, with different solutions ($H_2O$, 50% and 100% ethanol) and heart tissue sections of mice. The results showed that the specimen structures and components were kept intact in 2-hour perfusion, reflected by a similar structure of images captured at the beginning and the end of perfusion (see S6A Fig in S1 File). We repeated 6 rounds of experiments and pooled the data on the mean intensity. Comparison of the mean intensity at the beginning and end of perfusion led to a highly statistically significant

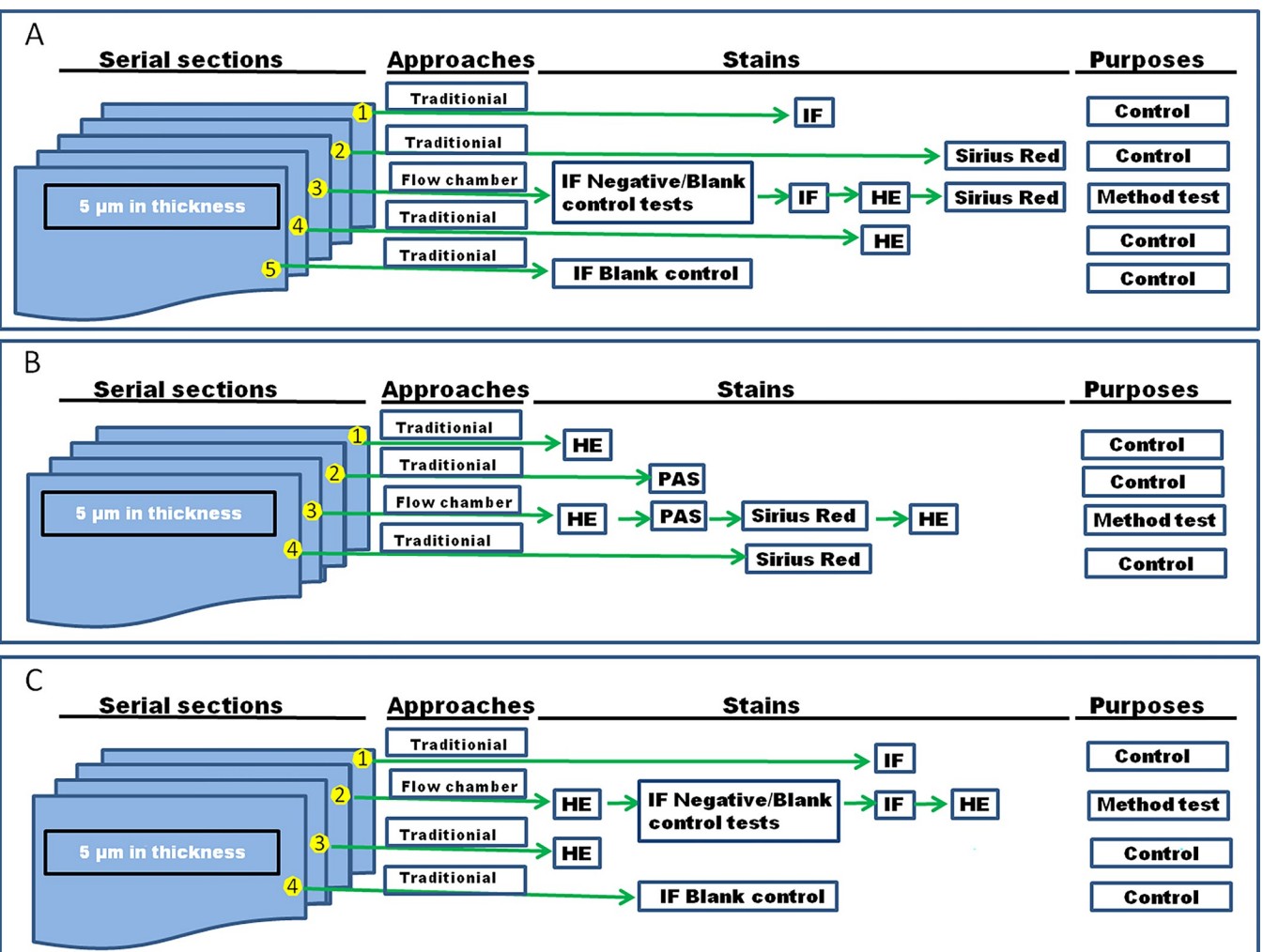

**Fig 2. Schematic flowchart of experiments. (A)** The diagram of immune fluorescence for Human Fc of Revacept, coupled with HE and Sirius red stains. (**B**) The protocol schema of HE stain, coupled with PAS and Sirius red stains. (**C**) The protocol schema of HE stain, coupled with immune staining (IF).

correlation (see S6B Fig in S1 File; N = 18 pairs, Pearson's r = 0.9836, P < 0.0001 in Supporting Information). No tissue fragments of the histological section were seen lost or washed away (see S6A Fig in S1 File) in 2-hour perfusion with respect to the images at the beginning.

The selected flow rate of 1 ml/min was based on the optimization of the trial tests on 1-hour perfusion of the heart tissue. The criteria for the optimization include good preservation of intact tissue following perfusion, relatively short turnaround time, small solution volume for staining, and good coordination of the relevant apparatuses (e. g., tubings size/length, containers of staining solutions).

## Validation of multi-stain combination

Blending multi-stains would provide additional information to pathologists and could help make diagnoses efficiently. A highly accurately merged digital image is premised on the precise site orientation of the section and non-drifting images acquired during flow chamber staining. To check whether the FCS flow state might impact the imaging and blending, we stained heart sections with hematoxylin in the flow chamber. The cell nuclei in the hematoxylin-stained

section, which became blue when contacted with slightly basic tap water, and red when contacted with an acid ethanol solution, were computationally registered and blended with the blue-nucleus images overlapping the red-nucleus images. We observed that the red and blue colors of Hematoxylin-stained nuclei displayed at the different pH values were well corresponded (see S7 Fig in S1 File). The good colocalization in merged images between two colored nuclei was reflected with a nearly 100% overlap rate (mean ± SEM, 0.99673 ± 0.0082) from 8 rounds of measurements, which verified the feasibility of the image blending technique in FCS.

## IF coupled with HE and Sirius red stains

It is common practice to start a new setup from a familiar field and simple style. For tests of the method, and 1) to know whether Revacept (GPVI-Fc, GPVI conjugated humanized Fc) binds to heart tissue, 2) to understand where the binding exists, and 3) to determine which structures are bound, we performed stains in heart tissue with FCS (refer to Fig 3A). In IF stain for human Fc of IgG, a negative control test was first performed with perfusions of blocking medium, anti-human Fc biotin-conjugated antibody, and avidin labeled with FITC in order (see Fig 3A). On the observation of the negative signal, the section was further objected to perfusion and/or incubation with Revacept and subsequently staining solutions in the same way as the negative control test but without antibody application (blank control). Without sight of signal on the section, IF staining was continued and finished with perfusion of blocking medium, antibody conjugated with biotin, and avidin labeled with FITC. As a result, a strong signal of green fluorescence was observed after perfusion of Revacept and biotin-labeled antibody, and nearly no signal could be seen in the negative or blank control tests in the same single section (see Fig 3A). The result verified the feasibility of the staining method in the tissue tested. To localize the signal within tissue structures, we additionally performed the stains of HE and Sirius red on the IF-stained sections by FCS. The re-staining effects displayed distinct tissue structures and strong color contrast. With the aid of the image blending technique, we found that Revacept merely binds to collagen (see Fig 3C). Similar effects were shown between the stains made by FCS and traditional histological procedures for IF, HE, and Sirius red stains in adjacent sections (Figs 2A and 3D), which demonstrates that the staining outcomes by FCS are independent of the destaining processing or flow chamber flow.

To test the feasibility of the method on other tissues, we simplified the protocol with initial immune staining (IF) and the subsequent HE staining mode—the potentially widely used protocol to localize and visualize the IF signals in HE-stained sections. The tissues used were the lung, liver, and kidney, and the targets to be detected included mouse CD31, hemoglobin, and CD45. Images generated by IF and HE with FCS and traditional staining procedures in the adjacent sections were demonstrated for comparison (S8-S10 Figs in S1 File). A similar staining effect was also found between FCS and traditional fashions in those tissues.

## HE coupled with histological special stains

Information in histology sections is usually visualized by different staining techniques. We chose esophagus tissue and three different stains—HE, PAS, and Sirius red stains to demonstrate the framework of multi-stains by FCS, as these stains are jointly used for inspection of possible morphological modification, which might contribute to large neck in the hyperthyroidism case. As summarized in Figs 2B and 4A, we revealed the method that can be used to perform HE stain along with histological special stains with a discoloring process in between (Fig 4C). The PAS or Sirius red-stained slides after discoloring of HE showed clear structures, accurate targets, clean background, and intact tissues (Fig 4A). There were no rips or tears in

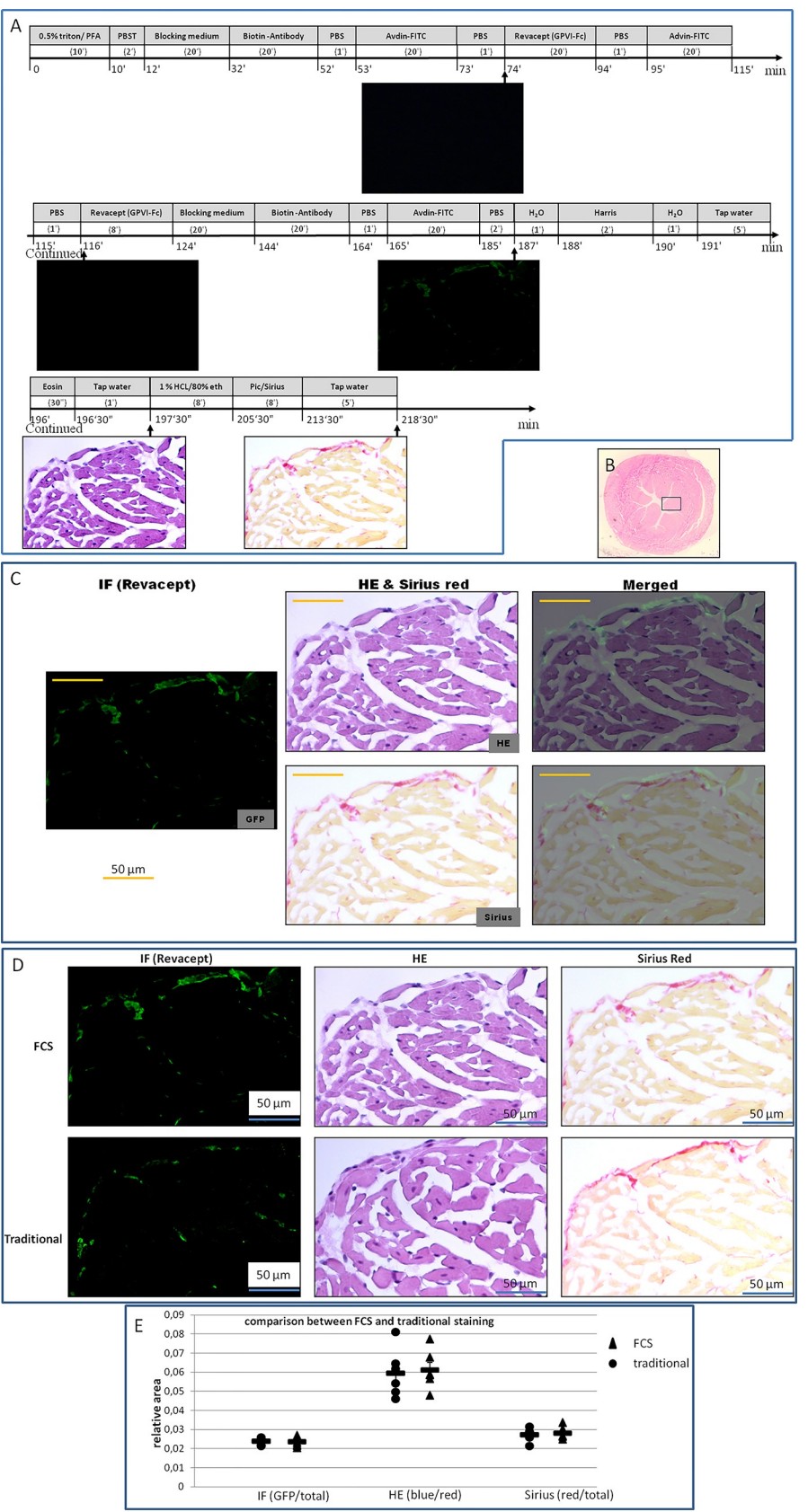

**Fig 3. Flowchart protocol of flow chamber staining for IF, HE, and Sirius red stains in heart tissue, and blending of the observed images. (A)** Flow chamber staining steps and time points for image acquisition. Numerals in

parentheses indicate the duration (minutes or seconds) of the corresponding agent perfusion (or incubation for >5 minutes of the span). The incubation was performed by a halt of perfusion after the chamber was filled with the corresponding agent. (**B**) The targeted region for inspection is a papillary muscle, indicated by a black inset in heart tissue. (**C**) Combination of images generated by different stains. The combination demonstrates IF-revealed-Revacept distribution in the HE or Sirius red-stained section. Codistribution of Revacept and collagen displayed in Sirius red was observed in heart tissue. (**D**) Images generated by different stains of FCS and traditional staining procedures in adjacent sections were demonstrated for comparison. (**E**) In terms of the relative areas, the comparison between the conventional and FCS, with Tukey HSD of one-way ANOVA, resulted in no significant statistical difference (N = 6 per group, for all, P = 0.537–0.829).

the re-stained sections found. The manifestation demonstrated in FCS showed no difference from that of the traditional stains (Figs 2B and 4D).

To extend FCS application to other tissues, namely, the lung, liver, and kidney–the most used tissues in the laboratory of biomedicine and clinic, we succeeded in staining those single sections by FCS (S11A-S11C Fig in S1 File). For the relatively fragile tissue (e.g., mouse brain), we also performed HE, PAS, and Sirius red stains by FCS. The outcome of the stains was videoed (S12 Fig in S1 File and Representative S4–S7 Videos).

## HE coupled with immune stains

To confirm or define the suspected components or structures found in the HE stained section, we first stained the brain tissue with HE by FCS, focused on the regions of interest, and then stained it with IF for mouse CD45 (Figs 2C and 5A). After discoloration of HE, the stain of IF showed that there was nearly no background noise. The cell structures shown by FCS were clear and the positive signal was strong in comparison with the traditional stains in the adjacent sections (Fig 5C). Similar results were also found in heart and brain tissues with alternative antibodies (S13 and S14 Figs in S1 File).

## Accuracy and comparison between FCS and traditional staining

Accuracy is the closeness of agreement between the tested results and accepted reference values [16,17]. Staining results with traditional (one-section-one-stain) histological procedures are widely used and accepted for morphological analysis. The degree of closeness between the outcomes by FCS and traditional staining results was tested by comparison of the targeted relative areas. To minimize possible biases of staining processing and evaluation, we repeated those experiments 5 times more in the same region for each tissue in different animals. The comparison between the traditional and FCS in terms of relative areas resulted in no significant statistical difference (N = 6 per group, for all, P = 0.085–0.829, see Figs 3E, 4E and 5E), which indicates histological stains with FCS are feasible.

The degree of closeness was also tested between FCS and traditional staining by assessing the correlation. We pooled the data from FCS and traditional procedures in the heart, esophagus, and brain, and compared the values determined in the corresponding regions of interest on the adjacent sections for each kind of stain. The results, as indicated in Fig 6, demonstrated a statistically significant correlation.

The similarity in the morphological features of the multi-stained images generated by FCS to those of images generated with traditional staining procedures in the adjacent sections, confirms that staining outcomes by FCS are accurate.

## Reproducibility and repetitive evaluation of signal relative areas

The method must be reproducible and not be affected by day-to-day variation [16,17]. The sections from the same regions, namely the papillary muscle for the heart, the cervical segment

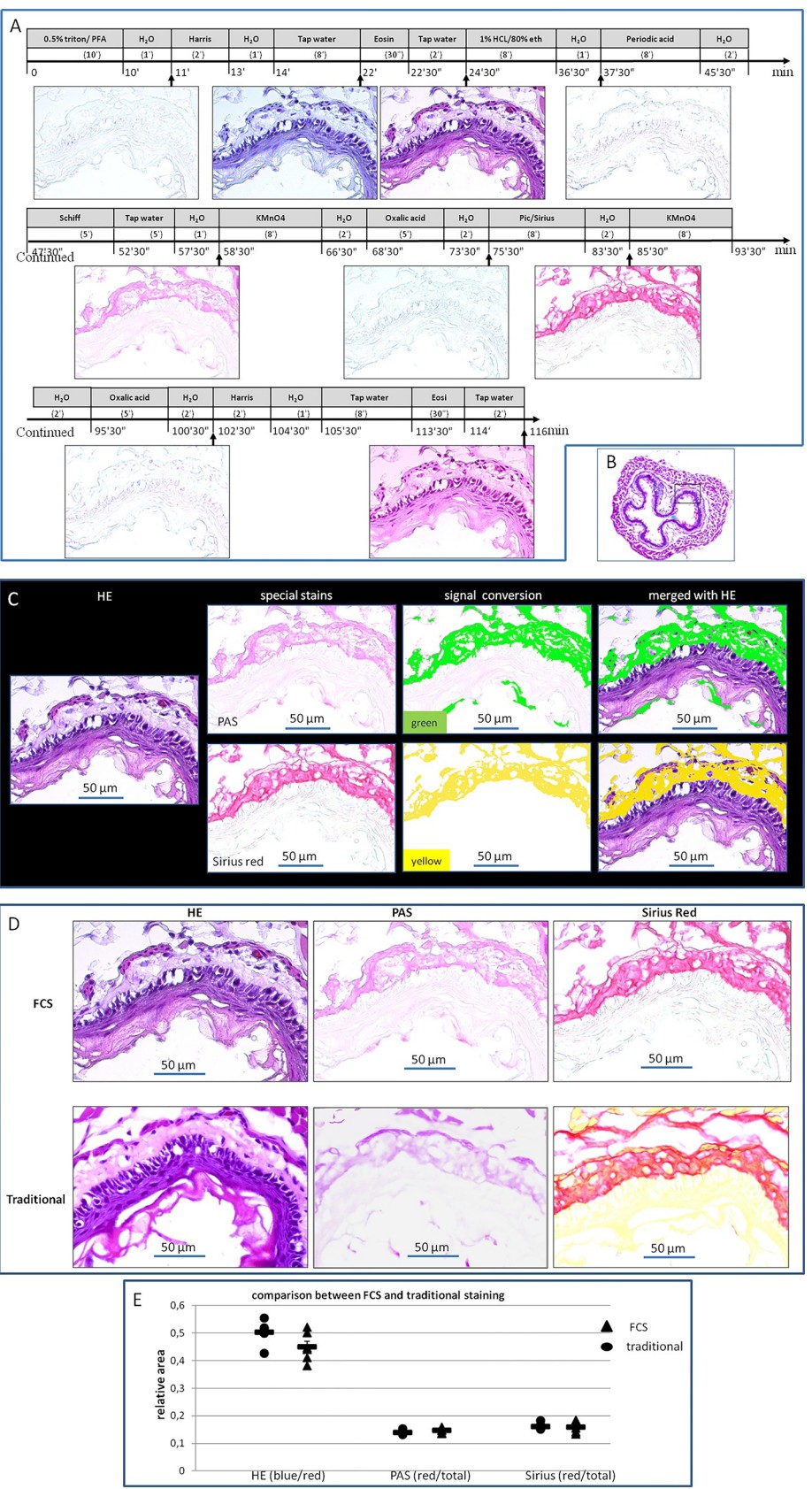

**Fig 4. Flowchart protocol of HE, PAS, and Sirius red stains by FCS, and the outcomes observed in esophageal tissue. (A)** Flow chamber staining steps and time points for image acquisition. Numerals in parentheses indicate the duration (minutes or seconds) of the corresponding agent perfusion (or incubation for >5 minutes of the span). The incubation was performed by a halt of perfusion after the chamber was filled with the corresponding agent. **(B)** The targeted region for inspection is the esophagus wall in the cervical segment, indicated by a black inset in the esophagus cross-section. **(C)** Blending images generated by HE, PAS, and Sirius red stains. The blending demonstrates that PAS-positive areas (green color) were mainly distributed in the outer membrane and to less extent, in muscodermis. The collagen (yellow color) revealed by Sirius red had a colocalization with PAS in the outer membrane. The Green or yellow color was converted from the red of staining targets in PAS or Sirius red, to avoid confusion. **(D)** Images generated by different stains of FCS and traditional staining procedures in adjacent sections were demonstrated for comparison. **(E)** In terms of the relative areas, the comparison between the conventional and FCS, with Tukey HSD of one-way ANOVA, resulted in no significant statistical difference (N = 6 per group, for all, P = 0.085–0.535).

for the esophagus, and the brain dorsal cortex at Bregma 0 mm (Figs 3B, 4B and 5B) of native animals (6 rounds, see S1 Table in S1 File), should return similar outcomes if reassessed at a later day with the same staining procedure. To determine the repeatability of the method, we repeated the measurements of the targeted relative areas in each kind of tissue and stain, and the results, as shown in Table 1, were in high similarity among 6 runs of measurements. All six runs were compared using one-way ANOVA and Tukey post-test for multiple comparisons of SPSS, resulting in, for all, p values of > 0. 64506 and thus no significant difference between the measurements on different days. The variation coefficient for 6 runs on different days was in the arrangement of 0.055–0.207 in terms of relative areas, which reflects a rather high reproducible procedure of the technique over time.

**Conclusions.** The new features of FCS non-seen in the traditional staining fashion including quick switching between de-staining and re-staining, real-time inspection and snap capture of multiple staining phenotypes, and efficient blending of multiple stains, do not affect the results of both histological and immune stains. The results also imply that FCS may be a valuable diagnostic strategy.

## 3. Discussion

The image blending of different histological and/or immune stains in the same tissue section is currently difficult with a traditional staining procedure. Our approach entirely overcomes the difficulty, allowing for multiple stains in the same tissue section and combination of the stains with ease, while the staining outcomes are real-time observed and snap captured, and not impacted by the alternative modality. The success of this approach has been validated using heart, esophagus, brain, lung, liver, and kidney tissues, involving immune stain (IF) for Human IgG, CD31, hemoglobin, and CD45 and three different histological stains—HE, PAS, and Sirius red. By blending multiple stains of FCS, the tissue-associated characteristics can be visualized simultaneously, to trace the positions of the signals observed in IF or to make out the natures of the suspected structures displayed in HE. For example, Fig 3C displays combinations between IF and HE or Sirius red. A combination of IF with HE indicates that Revacept detected in IF exists in the extra-cellar space or internal membrane of the heart papillary muscle, and the combination with Sirius red reveals that Revacept merely binds to collagen in the extra-cellar space or internal membrane. For another example, Figs 4C and 5C demonstrate blending between HE and PAS, Sirius red, or IF. HE enables easy differentiation of cell nuclei and cytoplasms and confers general structures, while PAS stain provides contrast to glycoproteins or mucins, Sirius red stain to collagen, and IF to specific compositions. The stain combinations (Figs 4C and 5C) presented can be made on demand by simply changing or not the signal color into a divergent one for the outcome of each kind of stain, which broadens information by allowing a pathologist to view the same areas of a sample with all the stains.

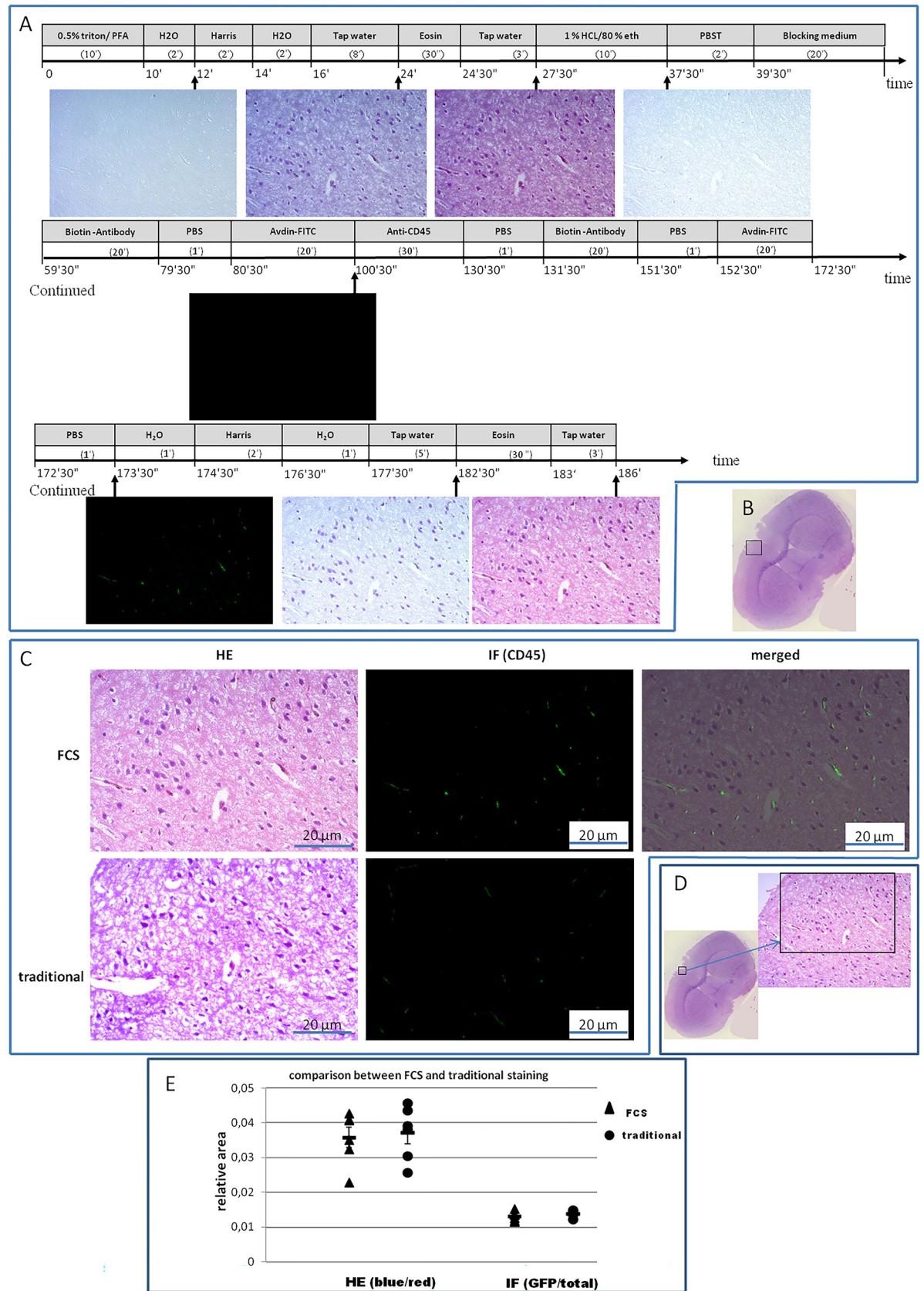

**Fig 5. Flowchart protocol of HE and IF staining by FCS, and the outcomes observed in brain tissue. (A)** Flow chamber staining steps and time points for image acquisition. Numerals in parentheses indicate the duration (minutes or seconds) of the corresponding agent perfusion (or incubation for >5 minutes of the span). The incubation was performed by a halt of perfusion after the chamber was filled with the corresponding agent. **(B)** Targeted region for inspection–dorsal cortex of brain at Bregma 0 mm, indicated by a black inset in the brain cross-section. **(C)** Blending images generated by HE and IF of FCS, and images from traditional staining procedures for comparison. The blending demonstrates the location of CD45-positive cells in the HE-stained section. **(D)** The local region of interest was magnified for inspection in (C). **(E)** In terms of the relative areas, the comparison between the conventional and FCS, with Tukey HSD of one-way ANOVA, resulted in no significant statistical difference (N = 6 per group, for all, P = 0.11769–0.7087).

By the features of re-staining and blending stains born in FCS, the archived-stained sections can be reused for the retrospective study. This spares extremely limited tissue samples, especially if dealing with biopsy material. Indeed, there are lots of similar situations in the clinic, such as a requirement of pathologically definite diagnosis or retrospection on just one archived HE-stained section available for suspected cases due to sampling source limitation. In addition, the problems such as the impractically long reiteration of preparation spent on dehydration, clear, coverslipping, imaging, decoverslipping, rehydration, etc. during restaining, as described [9,12,18,19], are eliminated. Therefore, a turn-around time in FCS is relatively shorter.

The flexible setup of FCS is not limited to applications on the multiple stains mentioned. Other staining protocols, like silver stains, in situ hybrid, etc. can also be performed with FCS. On the other hand, the flexible setup of FCS is not limited to the application of the frozen sections mentioned. Paraffin-embedded sections can also be used for stains by FCS, but it is necessary for the sections to be deparaffinated before assembling into the staining setup.

At the initial stage of FCS, as presented in this work, we emphasized single-target detection with conventional multiplexed histological and/or immune stains under a common-used

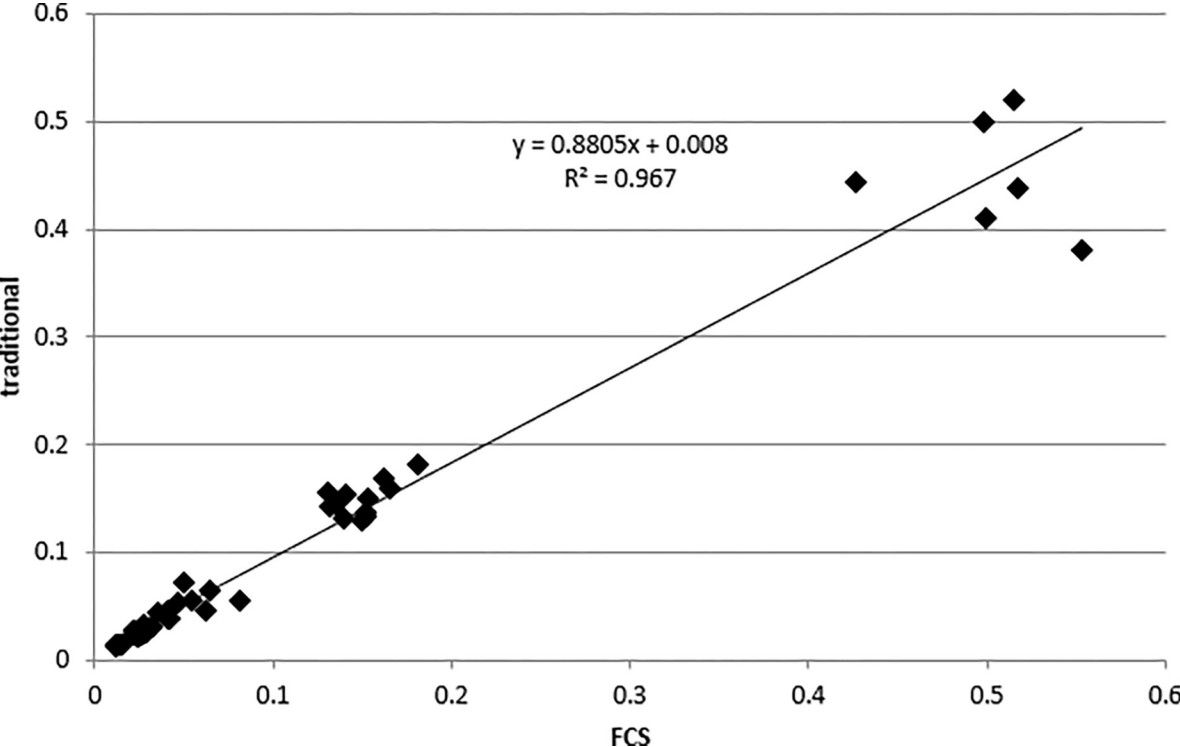

**Fig 6. Accuracy tested on target relative areas disclosed in stains of FCS and traditional fashion.** The statistical analysis resulted in a statistically significant correlation (N = 48 pairs, Pearson's r = 0.98336, P < 0.0001).

**Table 1. Relative areas of targets stained by FCS and coefficients of variation (CV) were obtained from six runs of measurements on different days.**

| Organs | Stains | Indexes | First | Second | Third | Fourth | Fifth | Sixth | CV |
|--------|--------|---------|-------|--------|-------|--------|-------|-------|-----|
| heart | IF | target related area | 0.025 | 0.022 | 0.025 | 0.021 | 0. 025 | 0.026 | 0.077 |
| | HE | blue/red | 0.050 | 0.054 | 0.046 | 0.065 | 0.073 | 0.062 | 0.173 |
| | Sirius Red | target related area | 0.029 | 0.026 | 0.029 | 0.022 | 0.027 | 0.032 | 0.125 |
| esophagus | HE | blue/red | 0.515 | 0.427 | 0.553 | 0.500 | 0.498 | 0.518 | 0.083 |
| | PAS | target related area | 0.135 | 0.131 | 0.152 | 0.132 | 0.141 | 0.140 | 0.055 |
| | Sirius red | target related area | 0.181 | 0.153 | 0.150 | 0.162 | 0.165 | 0.151 | 0.074 |
| brain | HE | blue/red | 0.023 | 0.043 | 0.035 | 0.041 | 0.032 | 0.041 | 0.207 |
| | IF | target related area | 0.013 | 0.011 | 0.012 | 0.015 | 0.014 | 0.012 | 0.107 |

The coefficients of variation for the 6 runs range from 0.055 to 0.207. Multiple comparisons with Tukey HSD of one-way ANOVA resulted in no statistical significance among the groups of measurements on various days for each organ (N = 2–3 per group, for all, $P > 0.64506$).

microscope. Detection for multiplex targets with multiplexed antibodies, one of the most important advantages in the field of morphology, which was not tested in the current work, can be also carried out to visualize multi-proteins in a single section with FCS. But optimization for antibodies, incubation/perfusion period, destaining, coloration, etc. should be performed beforehand.

FCS staining can normalize the stain quality with automatic control of the flow speed, duration, and environment temperature, allowing for its consistency and removing variations (due to, for example, the manual histochemical staining). This is certainly the desired feature and will help to improve the consistency of diagnoses.

The advent of digitized images in pathology has propelled this traditional field into what is now described as digital pathology [20]. To conform to times of digital pathology, novel staining and imaging platforms are required in concert. Flow chamber staining is one of these platforms by providing researchers with the tools for live observation of staining processes and results. The real-time visualization of the tissue staining process and results, which is very important for the solution of the most challenging problems in telediagnoses, sometimes is required. Video streams of the staining process and digital images can be shared in real-time. FCS has paved the way for this development, thus bridging physical distance (telepathology) between local hospitals, colleges, teachers, and students, and between home and workplace. The staining processing can also be videoed, backed up, and replayed for off-lined pathologists for suspected cases.

In addition, the staining technique of FCS is simple and consistent with the diagnostic workflow currently used by pathologists, which enables multistaining without disturbing the current laboratory workflow and has the potential to become a common supplementary tool for digital histopathology. On the other hand, the mistakes that might occur during staining can be real-time observed and mended by adjusting the protocol immediately.

Some factors, which affect the staining outcomes, should be paid attention to before the application of FCS. The universal factors, which affect the staining procedure of conventional staining fashions, will also affect the staining outcomes of FCS, such as temperature and light since the principle behind them is the same. However, the special factors on FCS are the solution flow rate, bubble formation in the chamber, the thickness of coverslips used for tissue section support, and imaging resolution at the higher magnification (e.g., 40x objective lens). For the imaging resolution, traditional stains of HE display better resolution or quality especially under higher magnification in comparison with FCS. But FCS gives out stronger signals in immunofluorescence (see S8-S10 Figs in S1 File for reference).

A successful application of the method mainly depends on 1) section integrity preserved throughout the staining, 2) the effectiveness of bleaching, and 3) immediate removal of air bubbles during perfusion.

## Section integrity

To prevent the tissue of sections from detaching glass coverslips and to preserve the adhered tissue architecture intact during a long time of treatment with staining and bleaching, cryosections are at best mounted on fresh poly-L-lysine coated coverslips and post-fixed in 4% PFA before histological processing.

## Bleaching techniques

Bleaching techniques are important to remove staining colors that otherwise interfere with the following stains. The best bleaching procedure should be to remove all the visible stains while retaining tissue integrity and morphology, not cause detachment of tissues from the coverslips, and not affect the antigenicity when preparing immunohistochemistry or IF assay. Long time, more frequency, and higher concentration of using bleaching reagents would be prone to induce section-detaching. To optimize various factors involved in the bleaching procedure, we made several tests by trial and error to determine the best outcome and established rapid and effective bleaching methods in cryo-sections. Those bleaching methods are easy to be incorporated into both the conventional and FCS staining protocols. However, the concentration and bleaching duration of the bleaching solution can be varied based on the section thickness. Ammonia and NaOH show good effects on the bleaching of the color products of PAS but easily induce section-falling. The antibody used for this study yielded promising results but different antibodies demonstrate various qualities, and therefore, continued optimization may be warranted for this procedure since the bleaching reagents, their concentration, and bleaching span may have an impact on results for some specific antibodies. The slices after decolorization can be selectively stained as required, and the original staining can be restored after satisfactory results are obtained.

## Removal of air bubbles

Air bubbles, as found in the flow chamber, can be very detrimental to staining. Identifying the reasons is the first step to eliminating them. Air bubbles inside fluidic channels usually appear when 1) one or several fittings are leaking, 2) the solutions are changed, and 3) the gas contains gaseous forms in the perfusion liquids during fluidic experiments. There is no universal solution that can guarantee a bubble-free experiment. But what we did, which has greatly improved the experiments, can serve as a reference. Here are listed some tips– 1) to ensure that no fitting is leaking, 2) to balance the fresh perfusion solution at room temperature for 10–15 min before use to release the potential gas contained, 3) to switch to 100% ethanol for a change when the bubbles present in the fluidic channel, and 4) to apply a debubbling device custom-made (Fig 1) that can be additionally integrated to the fluidic setup.

## 4. Limitations

Despite successful stains that have been made with this technology, the current technique has at least five weaknesses, inclusive of uncommonly used coverslips, a lack of a temperature controller, and low efficiency for high throughput of staining sections, limiting of image resolution under 40x objective lens, and occasional difficulties with sections of multiplexed targeted regions. 1). The uncommon coverslips (40 mm in diameter) used for the present study would

need serial special devices for coating, staining, storing, etc. Fortunately, more recently, com-mercially-produced flow chambers (Applied BioPhysics, Bioptechs, C & L Instruments, Inc., Fluxion, Glycotech Corporation, ibidi, Provitro, Stovall Life Science, Inc. Warner Instruments, Inc.) in a variety of geometries are available that potentially overcomes these weaknesses. For the detailed sizes of applications, refer to the references [21–34]. 2). There was no temperature controller integrated into the setup, the environment, and perfusion solution temperature may affect the results. But we made the experiments with the environment temperature of 23˚C. The staining which requires a divergent temperature would be modified in protocol accord-ingly. 3). FCS is very highly efficient for scarce samples, with intention of intensive investiga-tion. Application of FCS is not encouraged for high throughput of staining sections. 4). Without the application of a clearing agent (e.g., xylol) and mounting medium of the tradi-tional staining process, the setup of FCS compromises the visual resolution of the images to some extent although it does not affect the diagnosis. 5). Sometimes, it would be problematic with a section of multi-targeted regions, when stained with FCS since only one target is nor-mally focused on in FCS. To improve the situation, we suggest changing objective lenses to various magnifications to relocalize the desired region of interest while moving the flow cham-ber for orientation.

## 5. Materials and methods

This work, inclusive of tissue processing, cutting, staining, etc. was completed in a laboratory with a room temperature of 23˚C.

### Coverslip preparation

Coverslip preparation includes cleaning and coating steps. For cleaning, 1) take coverslips (40 mm coverslips, Cat# 40–1313, Bioptechs Inc, USA) onto a beaker, fill the beaker with distil water, sonicate (Bandelin Sonorex, Cat# RK 100, Brandelin Electronic, Germany) the cover-slips for 5 min and swirl occasionally; 2) pour off the water, add 0.5% (vol/vol) Dr. Weigert (Laboclean FM, Cat# CLP0.1, Carl Roth, Germany) to cover the cover glasses, sonicate it for 15 min and swirl the beaker occasionally; 3) pour off the Dr. Weigert solution, fill the beaker with distil water; 4) sonicate the coverslips for 5 min, swirling occasionally, pour off the water and fill the beaker with fresh distil water; 5) repeat the steps of the sonication, and pouring off and filling water for 5 times; 6) pour off the water and fill 100% ethanol; 7) sonicate the cover-slips for 5 min, swirling occasionally, pour off the 100% ethanol and fill the beaker with fresh 100% ethanol; 8) repeat the steps of the sonication, and pouring off and filling 100% ethanol for 3 times. For coating, 1) separate the coverslips to let them air dry for 30 min; 2) cover the coverslips (300 μl) with 0.01% (wt/vol) poly-L-lysine solution (Cat# P8920, Sigma); 3) rock gently to ensure even coating of the surface during incubating; 4) after the Incubation of 30 min at room temperature, remove the solution by aspiration and allow to dry at least 2 hours before use. We summarized the steps in a video (refer to S1 Video).

### Animals and specimen preparations

All animal experiments were performed in accordance with Directive 2010/63/EU and approved by the Government of Upper Bavaria, reference number: ROB-55.2-2532.Vet_02-19-69 and ROB-55.2-2532.Vet_02-16-115, based on prior evaluation of animal study plan design and group sizes by the certified bio-statistician Dr. Peter Klein. The description of all procedures involving animals was done according to the ARRIVE Guidelines in reporting in vivo experiments [35].

The mice which were euthanized by cervical dislocation in deep anaesthesia (Ketamin 150 mg/kg and Xylazin 15 mg/kg) are summarized in S1 Table in S1 File on the strain, gender, age, body weight, and source. The hearts, cervical esophagi, lungs, livers, kidneys, and brains were excised from the euthanized animals. After removal of the around fat tissues, the samples were washed in ice-cold saline, trimmed the samples to middle sections of 5 mm in thickness with two cross cuts (refer to Figs 3B, 4B and 5B & S7-S10 Figs in S1 File), and fixed in 4% paraformaldehyde at 4˚C temperature for 24 hrs. The fixed samples were kept in 30% (w/v) sucrose (prepared in PBS) for at least 4 hrs at 4˚C until used for OCT embedding (Optimal cutting temperature compound, VWR Chemicals, Leuven, Belgium).

## Embedding, sectioning, and collection

For OCT embedding, the trimmed tissue blocks were incubated in OCT for 6 hours, with 3 changes of fresh OCT under sonication at room temperature after incubation in 30% (w/v) sucrose, and snap-frozen in a dry ice/isopentane bath. Consecutive 5 μm thick sections were cut using a Leica microtome—CM1850 cryostat (temperature, − 20˚C, Leica Biosystems, Buffalo Grove, IL, USA) and mounted on adhesive slides (Poly lysine slides, Cat#63700-w1, Roth, Germany) for traditional staining and the coated coverslips for flow chamber staining (refer to Fig 2), as reports before [16,36]. Five serial sections were cut for heart tissue (Fig 2A) and four serial sections for the other tissues (Fig 2B and 2C).

## Flow chamber setup and flow chamber staining (FCS)

The coverslips mounted with sections were assembled into the flow chamber device including an FCS3 chamber and microaqueduct slides (Cat# 21-060319-3, Biotechs Inc, USA). Care was taken to precisely place the section within the borders formed by the silicone gasket slit in the flow chamber. The flow chamber device was then connected via tubings (Silicone tubing, Cat# 68-015-075, VBM Medizintechnik GmbH, Germany) with the liquid reservoir on one end and with the perfusion pump (Pump 11, Cat# 70–4504, Harvard Apparatus, USA) on the other end (Fig 1 and S5 Fig in S1 File). For convenience, we created a custom-made holder for the flow chamber setup from a dish lid (S5 Fig in S1 File). The staining phenotypes were observed under a microscope (Axioscope A1, Carl Zeiss, Germany). The microscope is equipped with a fluorescence apparatus and a camera–coupled Zeiss imaging system. An upper gasket (FCS 2&3 Gasket 0.5 mm, 14x24, Cat# 060319-2-0719, Biotechs Inc, USA) used for present flow experiments is 500 μm in thickness and a slot of 14x24 $mm^2$ (an observable window). The flow chamber perfusion was made by pump withdrawal (velocity: 1 ml/min). We summarized the steps in videos (refer to S2 and S3 Videos).

Before FCS, the setup was validated with initial perfusion fixation of 4% formaldehyde for 10 min, followed by H2O, 50%, and 100% ethanol respectively for 2 more hours on each fresh section of heart tissue (S6A Fig in S1 File). The regions for observation in the sections were randomly selected and photographed with an x 20 objective lens at the ends of the fixation and the 2-hour perfusion. The mean intensity of the images captured was determined using the software–Image J.

## IF coupled with HE and Sirius red stains and HE coupled with special stains or immune stains

IF staining is the most frequently used technique to detect a wide variety of antigens in different types of tissue preparations, which allows for excellent sensitivity and amplification of signal in comparison to immunohistochemistry. Localizing the antigens and getting acquainted with the related structures unveiled by HE stain would improve diagnosis. We selected

**Table 2. Main reagents used for FCS and traditional staining.**

| REAGENTS | SOURCE | IDENTIFIER |
|---|---|---|
| Phosphate-Buffered Saline (PBS) | Corning Product | Number 21-040-CV |
| Paraformaldehyde (PFA), working conc: 4% in PBS, pH 7.4 | Carl Roth, Germany | Cat# 0335.3 |
| Tween 20, working conc: 0.5% in PBS (PBST) | Carl Roth, Germany | Cat# 9127.1 |
| Triton X100, working conc: 0.5% (v/v) | Carl Roth, Germany | Cat# 3051.3 |
| Bovine serum albumin, biotin free (BSA) | Carl Roth, Germany | Cat# 0163.4 |
| Mouse serum | Dako, Denmark | Cat# X0910 |
| Rat anti-Mouse CD45 (anti-CD45), working conc: 4 µg/ml | BD Pharmingen, BD Bioscience | Cat# 550539 |
| Rat anti-Mouse CD31, working conc: 1:20 | Dianova, Germany | Cat# DIA-310 |
| Biotin conjugated Hemoglobin Antibody, working conc: 20 µg/ml | ILSBio,LifeSpan BioSciences, Inc | LS-C212174 |
| Revacept, working conc: 30 µg/ml in PBS | Orpegen GmbH, (now glycotope GmbH) | Batch Cat# PR-15-DP-03 |
| Biotin conjugated Mouse Anti-Rat IgG (Biotin-Antibody), working conc: 1:500 | Thermo | 212-065-168 |
| Biotin conjugated Mouse Anti-Human Fc(Biotin-Antibody), working conc: 8 µg/ml | Jackson Immunoresearch, USA | Cat#209-065-088 |
| Streptavidin conjugated FITC (avidin-FITC), working conc: 1:500 in PBS | Jackson Immunresearch, USA | Cat# 10-22-23 |
| Harris hematoxylin (Harris) | Sigma | Cat# HHS80 |
| Glycerol, working conc: 20% | Carl Roth, Germany | Cat# 7533.1 |
| Eosin G | Carl Roth, Germany | Cat# 7089.1 |
| Sirius red F3B /Direct Red 80 | Fluka, Germany | Cat# 43665 |
| Oxalic acid, working conc: 1% (w/v) | Carl Roth, Germany | Cat# T113.1 |
| $KMnO_4$, working conc: 0.5% (w/v) | Carl Roth, Germany | Cat# P752.1 |
| Hydrochloric acid 5M | Carl Roth, Germany | Cat# 1E2C.1 |
| Saturated aqueous solution of picric acid | Sigma | Cat# P6744-1GA |
| PAS staining kit | Carl Roth, Germany | Cat# HP01.1 |
| Ethanol | Carl Roth, Germany | Cat# K928.4 |
| Xylol | Carl Roth, Germany | Roth, Cat# 6640.1 |
| Kanadabalsam | Carl Roth, Germany | Cat# 8016.2 |
| Blocking medium: 1% BSA and 1% mouse serum in PBS. | | |
| Picro/sirius Red Solution (Sirius): 0.5 g of Sirius red F3B solved in 500 ml of Saturated aqueous solution of picric acid. | | |

Abbreviated names between parentheses are cited in Figs 3, 4 and 5.

Revacept (GPVI-Fc) binding tests by FCS. We first carried out IF stain by FCS, and then HE and Sirius red stains (Fig 2A, and for more details, see Fig 3A).

On the other hand, HE stain is routinely employed for the acquisition of general information on the status of cells and tissues. For the suspected regions of interest, special stains or IF are carried out to improve the accuracy of diagnosis. Thus we first performed HE stain by FCS, and then PAS and Sirius red stains, or IF (Fig 2B and 2C).

The Flow chamber was perfused at a velocity of 1 ml/min with 4% paraformaldehyde for tissue fixation, followed by various corresponding solutions for HE, PAS, and Sirius Red stains, or IF (see Figs 4A and 5A for more details). The phenotypes of stained tissues were captured under a bright field of the microscope. The perfusion solutions involved in fixation, washing, blocking, staining, antibody binding, destaining, and restaining were described in Table 2.

### Image acquisition and image analysis

Images were acquired using the Zeiss microscope and imaging system using a 20x Objective and recorded with a 2560 x 1920 pixel resolution and JPEG mode, or videoed with an extended camera (Canon Power shot A650 IS). For IF stained sections, green fluorescence proteins were

inspected and photographed in a fluorescence channel (488 nm excitation and 543 nm emission).

HE, PAS, and Sirius red-stained slides were examined and photographed under a bright field. The images were acquired under a lighting condition except for focusing on each new visual field. The lighting condition includes exposure time, 30.0 ms; color cold, 0.3; color saturation, - 0.2; light strength, - 0.24; and color contrast, 5.92.

For image analysis, the JPEG images of unstained sections were opened with ImageJ software, and the mean intensity was determined.

The relative area of targets (signals) is the ratio between the target (signal) area and the involved (total) area in a stained section. The target areas and the involved (total) area were determined with Photoshop Software (Adobe Photoshop V5) as done before [16,37–42].

## Stain blending

A stain mixture is generated by combining two or more stains in the desired tissue areas or signal colors at controllable ratios (see Figs 3C, 4C and 5C). In other words, the newly generated stain can be tuned on demand by simply converting signal colors and changing the color intensity, thus making the different stain combinations more or less pronounced. Figs 3C, 4C and 5C demonstrate several such stain combinations. For stain blending, the JPEG images to be merged were opened with Photoshop (Photoshop Element 6, Adobe). First, select and copy the whole image outline of one stain, and then paste or overlap the image of another stain. Finally, adjust the light transparency degree (we used ~50%) and a merged image with different color signals was shown. For the combination of 3 stains, select and copy the whole image outline of the third stain, and then paste or overlap the image on the merged image. To highlight a specified signal or avoid confusion on signal colors, the signal color could be converted into a desired one by a selection of the target signal color and filled with the desired color (Fig 4C). In such a case, a combined image can be reconstructed using pseudocolors and reveal different targets.

## Traditional immune and histological stains

For validation of FCS, as controls, we performed the corresponding conventional histological stains in fresh (unstained) adjacent sections of serials (Fig 2). The stains include IF, HE, PAS, and Sirius red stains, which are the most frequently used methods in histological laboratories (see S2-S4 Tables in S1 File).

For Revacept binding, the sections were washed, fixed, and incubated with Revacept. The incubated sections were ready for IF stain. We used the well-established protocol, which has been successfully applied in atherosclerosis and stroke in our laboratory [16,36,37,40–42]. The sections were immunostained for Hu IgG, using an avidin-biotin complex (ABC) method. As blank controls, one of the adjacent slides was incubated with Revacept and then subjected to IF staining, except that the biotinylated mouse anti-human IgG antibody was replaced by the same volume of PBS. As negative controls, the other adjacent slides without previous incubation of Revacept went through the same staining process. To see more detailed steps, refer to S3 Table in S1 File.

For IF targeted at CD45, hemoglobin, and CD31, similar procedures mentioned above were adopted. (For details, see S4 Table in S1 File).

For HE staining, the slides to be stained are indicated in Fig 2. The staining procedures comprised mainly 3 min Harris hematoxylin staining, bluing in tap running water, 40-sec Eosin staining, and coverslipping. For more details, see S2 Table in S1 File.

For Sirius red staining, the slides (Fig 2) were incubated with Pico-Sirius red solution for 15 min, and the staining ended with coverslipping. For more details, see S2 Table in S1 File.

For PAS staining, staining was carried out in 2 steps: glycol group targets converted to aldehyde groups by Periodic acid incubation and coloration of aldehyde groups with incubation of Schiff's reagent. For more details, see S2 Table in S1 File.

### Data analysis

Data were presented as mean ± SEM. The software SPSS (IBM Corp. IBM SPSS Statistics for Windows, Version 11.0., USA) was employed for the analysis of correlation or multiple comparisons of means. A $P < 0.05$ was considered statistical significance.

## Supporting information

**S1 File. Supporting information contains complementary information on materials and methods and supplemental results.**
(DOCX)

**S1 Video. Coverslip preparation.**
(MP4)

**S2 Video. Assembling of Flow chamber setup.**
(MP4)

**S3 Video. Tubing connection and pump setting.**
(MP4)

**S4 Video. HE staining in the section of the mouse brain by FCS.**
(MP4)

**S5 Video. PAS staining in the section of the mouse brain by FCS.**
(MP4)

**S6 Video. Pic/Sirius staining in the section of the mouse brain by FCS.**
(MP4)

**S7 Video. HE re-staining in the section of the mouse brain by FCS.**
(MP4)

## Author Contributions

**Conceptualization:** Zhongmin Li.

**Data curation:** Zhongmin Li.

**Formal analysis:** Zhongmin Li.

**Investigation:** Zhongmin Li, Kerstin Uhland, Andreas Reimann.

**Methodology:** Zhongmin Li, Silvia Goebel, Clara Wenhart.

**Project administration:** Zhongmin Li, Goetz Muench.

**Resources:** Zhongmin Li, Silvia Goebel, Clara Wenhart, Andreas Reimann.

**Supervision:** Goetz Muench.

**Validation:** Zhongmin Li.

**Writing – original draft:** Zhongmin Li, Kerstin Uhland.

**Writing – review & editing:** Zhongmin Li, Goetz Muench, Silvia Goebel, Kerstin Uhland.

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
