## [Decision Letter · Decision Letter 0]

16 Jan 2023

PONE-D-22-32143Flow chamber staining modality for real-time inspection of dynamic phenotypes in multiple histological stainsPLOS ONE

Dear Dr. Li,

Thank you for submitting your manuscript to PLOS ONE. After careful consideration, we feel that it has merit but does not fully meet PLOS ONE’s publication criteria as it currently stands. Therefore, we invite you to submit a revised version of the manuscript that addresses the points raised during the review process.

We look forward to receiving your revised manuscript.

Kind regards,

Kanhaiya Singh, Ph.D

Academic Editor

PLOS ONE

Journal Requirements:

Additional Editor Comments:

Although the Reviewers found this study of interest, they have raised significant concerns about the novelty of the study, clarification about several images and staining protocol. Please address these concerns in details.

Reviewers' comments:

Reviewer's Responses to Questions

**Comments to the Author**

1. Is the manuscript technically sound, and do the data support the conclusions?

Reviewer #1: Partly

Reviewer #2: Yes

2. Has the statistical analysis been performed appropriately and rigorously? 

Reviewer #1: Yes

Reviewer #2: Yes

3. Have the authors made all data underlying the findings in their manuscript fully available?

Reviewer #1: Yes

Reviewer #2: Yes

4. Is the manuscript presented in an intelligible fashion and written in standard English?

Reviewer #1: Yes

Reviewer #2: No

5. Review Comments to the Author

Reviewer #1: The authors presented an interesting approach for multiplex staining under the title “Flow chamber staining modality for real-time inspection of dynamic phenotypes in multiple histological stains”. The authors tried to address the current limitation of traditional immunohistology by developing a new technique that has multiple advantages compared to the traditional staining methods. Additionally, the authors reported that their technique is very useful in the situation of scarce tissue availability with real time imaging and recording. I have some concerns and comments:

Major concerns:

1- What is the novelty of this approach compared to the already available commercial products for automated immunohistology, especially the authors mentioned in the discussion “Fortunately, more recently, commercially produced flow chambers (Applied BioPhysics, Bioptechs, C & L Instruments, Inc., Fluxion, Glycotech Corporation, ibidi, Provitro, Stovall Life Science, Inc. Warner Instruments, Inc.) in a variety of geometries are available that potentially overcomes these weaknesses. What about the other factors that usually affect the staining procedure such as temperature and light?

2- One of the important aspects for evaluation of the new modality compared to the traditional methods is to assess the time used to complete the protocol of staining. Can the authors elaborate more on that compared to the traditional staining? Is it significantly shorter, longer or the same?

3- The authors mentioned “By the technique, staining processing can be videoed and made a backup for off-site pathologists, which facilitates tele-consultation or -education in current digital pathology”, I suggest adding a short video that can summarize the steps of the staining protocol to strengthen the authors claim and for reproducibility, especially that there is only schematic diagram in fig.1 and fig. S1without digital imaging of the setup itself.

4- In the abstract, the authors should state the gap and the current limitation in the conventional staining protocols.

5- In the Introduction, it is little redundant, and it can be more concise with supporting literature especially when the authors mentioned the disadvantages of multiplex staining. Additionally, the authors elaborated more on their methodology in the last paragraph which is more relevant to the method section.

6- In the results: the authors mentioned “Images generated by IF and HE of FCS and traditional staining procedures in the adjacent sections were demonstrated” Figs S4-S6, I noticed 1- that most of the sections used are not matching, were the sections used for both methods are identical? 2- Additionally, H&E is better resolution or quality in traditional staining. 3- Although the signal localization is close in IF but there are a lot of differences in IF stained sections in both methods, please check CD31 in Fig. S4, this would be problematic in specific small target regions assuming that the staining was done on identical sections for validation.

7- The authors should refine the results section and focus on their results only, some of the results text is more related to discussion.

8- The discussion section has a lot of redundancy similar to the results section. The authors should summarize their key findings and analyze how their results fit in with previous research of similar modalities and what is the added values. Also, some of the discussion is related mostly to suggestion and potential use of the modality which isn’t supported by data in the current study for example “FCS staining can normalize the stain quality with automatic control of the flow speed……….”.

9- The authors already mentioned a lot of limitation of FCS modality which can be limiting factor to adopt this technique. Did the authors test more than one antibody with this technique?

10- On of the most important advantages of the multiplex imaging and other new techniques is to target multi-protein or use multiple antibodies on the same section? Is it possible with the use of FCS?

Minor comments:

11- In figure 1 the authors demonstrated schematic diagram for the workflow, which is starting with pump, it’s little confusing with arrows direction. How is the syringe withdrawing from the reservoir, although the reservoir is positioned post or after the chamber?

12- Please add scale bars for all the images, especially supplementary figures.

13- The authors mentioned about supplementary videos, but I couldn’t find.

Reviewer #2: Manuscript Number: PONE-D-22-32143

Article Type: Research Article

Full Title: Flow chamber staining modality for real-time inspection of dynamic phenotypes in multiple histological stains

In this study, the authors introduced a new staining modality, flow chamber stain, which follows the current staining workflow but has additional features not found in conventional stains, allowing for (1) quick switching between destain and restain for multiplex staining in one section from routine histological preparation, (2) real-time inspecting and digitally capturing each stained phenotype, and (3) efficient staining. Hematoxylin-eosin (HE), Periodic acid–Schiff, Sirius red, and immunofluorescence for Human IgG, mouse CD45, hemoglobin, and CD31 were compared to its stains on microscopic pictures of mouse tissue (lung, heart, liver, kidney, esophagus, and brain). No substantial differences were found. The approach is accurate and reproducible in targeted stained section tests. Using the approach, immunofluorescence targets were easily located and visible structurally in HE- or special stained sections, and histological special or immune staining may be used to identify unknown or suspected components or structures.

However, the reviewer has some concerns:

Minor comments:

1) Figure 1 needs to be rectified. Not clear representation.

2) Abstract consists of few grammatical errors. Need thorough revision.

3) Page 4, “But the obvious…………..dropping”. Please reframe the sentence using appropriate words.

4) Page 4, line 104, replace the word ‘cover-slips’ with ‘coverslips’

5) Page 6, line 175-178, “Images of ………………..red-nucleus images. The sentence is not clear. Please use appropriate words with meaningful sentences.

6) Page 10, line 315, did you mean Tukey test? If yes? Please correct the word.

7) Page 12, line 362-363 please recheck the sentence and use appropriate words.

8) Page 19, table 2. Main reagents used for FCS and traditional staining, and Page 21, line 615. Please replace the word ‘Harri’s hematoxylin’ with ‘Harris hematoxylin’

9) In supporting Information- Table S3. Procedures of Immune Fluorescence stain for Human IgG (Fc specific). Did you mean ‘Immunofluorescence staining’, If yes, please correct the sentence everywhere accordingly.

10) Please maintain the same format (i.e.- Font size, theme and style) throughout the manuscript.

Major comments:

1. Please refer to the new existing similar technologies and define how the present work is different (pros and cons) from the existing technologies.

2. The work seems similar to the existing technologies e.g., Co-detection by indexing (CODEX). The author requires to justify the novelty of the present work with the similar existing technologies emphasizing the novelty of current work.

3. How did the author determine the flow rate to be 1 ml/min, and did they compare this value to other flow rates? If yes, please provide a brief description of the observation in the results section.

6. PLOS authors have the option to publish the peer review history of their article (what does this mean?). If published, this will include your full peer review and any attached files.

Reviewer #1: No

Reviewer #2: No

---

## [Author Response · Author response to Decision Letter 0]

8 Feb 2023

Additional Editor/Reviewer 2 Comments:

COMMENT 1:

In this study, the authors introduced a new staining modality, flow chamber stain, which follows the current staining workflow but has additional features not found in conventional stains, allowing for (1) quick switching between destain and restain for multiplex staining in one section from routine histological preparation, (2) real-time inspecting and digitally capturing each stained phenotype, and (3) efficient staining. Hematoxylin-eosin (HE), Periodic acid–Schiff, Sirius red, and immunofluorescence for Human IgG, mouse CD45, hemoglobin, and CD31 were compared to its stains on microscopic pictures of mouse tissue (lung, heart, liver, kidney, esophagus, and brain). No substantial differences were found. The approach is accurate and reproducible in targeted stained section tests. Using the approach, immunofluorescence targets were easily located and visible structurally in HE- or special stained sections, and histological special or immune staining may be used to identify unknown or suspected components or structures. 

RESPONSE:

Thanks for the inspiring comments. 

COMMENT 2:

Minor comments:

Figure 1 needs to be rectified. Not clear representation.

RESPONSE:

The comment is right. We have rectified it. Please see the updated Figure 1. 

COMMENT 3:

Minor comments:

Abstract consists of few grammatical errors. Need thorough revision.

RESPONSE:

Yes, the corrections have been made. Please refer to the red text in Abstract Section on Page 2.

COMMENT 4:

Minor comments: 

Page 4, “But the obvious…………..dropping”. Please reframe the sentence using appropriate words.

RESPONSE:

Yes, we have reframed the sentence using the appropriate words. Please see the red text on Lines 99-101, Page 4. 

COMMENT 5:

Minor comments:

Page 4, line 104, replace the word ‘cover-slips’ with ‘coverslips’

RESPONSE:

OK, we have replaced it. Please refer to the red text on line 106, Page 4.

COMMENT 6:

Minor comments:

Page 6, line 175-178, “Images of ………………..red-nucleus images. The sentence is not clear. Please use appropriate words with meaningful sentences.

RESPONSE:

Yes, we admit our mistake in the description. The correction using the appropriate words has been made. Please see the red text on lines 181-187, Page 6.

COMMENT 7:

Minor comments: 

Page 10, line 315, did you mean Tukey test? If yes? Please correct the word.

RESPONSE:

Yes, we have corrected it. Thanks! Please see the red text on line 322, Page 10.

COMMENT 8:

Minor comments:

Page 12, line 362-363 please recheck the sentence and use appropriate words.

RESPONSE:

Yes, we have corrected it according to the nice suggestion. Thanks! Please see the red text on lines 364-365, Page 12.

COMMENT 9:

Minor comments:

Page 19, table 2. Main reagents used for FCS and traditional staining, and Page 21, line 615. Please replace the word ‘Harri’s hematoxylin’ with ‘Harris hematoxylin’

RESPONSE:

Yes, the suggestion is right. The replacements can be seen on the red text in table 2, Page 19, and on line 625, Page 21.

COMMENT 10:

Minor comments:

In supporting Information- Table S3. Procedures of Immune Fluorescence stain for Human IgG (Fc specific). Did you mean ‘Immunofluorescence staining’, If yes, please correct the sentence everywhere accordingly.

RESPONSE:

Yes, a good reminder. We have checked and corrected the words in both the manuscript and Supporting Information. Please refer to S3 & 4 Tables in Supporting Information. 

COMMENT 11:

Minor comments:

Please maintain the same format (i.e.- Font size, theme and style) throughout the manuscript. 

RESPONSE:

OK, that is no problem.

COMMENT 12:

Major comments:

Please refer to the new existing similar technologies and define how the present work is different (pros and cons) from the existing technologies.

RESPONSE:

Yes, that is no problem.

Since the technique is a fusion method of traditional histological stains, including immune stains, and conventional flow chamber. Therefore, the closest technologies in similarity are the two methods. For the convenience of understanding, we extend these two methods to the new existing similar technologies.

 For the multiplexed stains in the new existing similar technologies, the techniques can be divided into antibody-based and artificial intelligence (AI)-based methods. 

The antibody-based techniques can be subdivided into 1) chromogen-labeled, 2) fluorescence-labeled, 3) Metal isotope labeled, and 4) DNA-barcoding- labeled multiplexing methods. 

The metal isotope labeled multiplexing methods include Imaging mass cytometry (IMC) and Multiplexed ion-beam imaging (MIBI).

DNA-barcoding-based methods include CO-detection by inDEXing (CODEX), Digital Spatial Profiling (DSP), and InSituPlex.

The methods mentioned were summarized in a table (please refer to the table below). We arranged the methods in decreasing order in the degree of similarity to FCS from top to bottom.

In the table, we sketch the general information on the relevant methods, emphasizing on the advantages and disadvantages. 

On the other hand, thanks to the considerable prompt, we have added some descriptions of the recent advancements related to the current work in Introduction section. The addition is marked with red. Please see the red text on Lines 83 and 84, page 3.

Table 1. Comparison of the different methods relevant to FCS.

 Purposes/application Sample /reagent preparation Staining types/process Detection modalities/synthesizing software Advantages Disadvantages Refs 

FCS Colocalization of tissue structures or cells among various stains on a single section Routinely prepared histological sections Histological stains; immune stains, or in situ Hybrid (to be tested) Upright/ Inverted microscope; Flow chamber; Software - Photoshop Multiplexed stains in a single section; easy for co-imaging across immune and traditional histological stains; Good compatibility with traditional workflow; Low cost Low multiplex; Additional set-up (flow chamber/pump, etc.); low throughput; Manual operation Current work

Traditional IF/IHC/histostains Morphological definition of tissue components, structures, or cells Routinely prepared histological sections Histological stains; immune stains; in situ Hybrid Upright microscopes; No special software needed Low cost; easy to handle; wide use. Good compatibility with traditional workflow One stain for a single section; lack of codetection of different stains 1, 2,3

Flow chamber Observation of interplay between coated reagents or cells under flow; Establishing invivo-like conditions for the cultivation of cells under flow Endothelial cells relevant to moving flow (e.g., vessels) or other coating substances Either non-staining or Fluorescence labeling Inverted microscopes. Flow chamber; No special software needed Straight-forward; easy to handle; mimicking invivo conditions; real-time observation of interplay between cell and coating substances Additional set-up (flow chamber/pump, etc.); limited to flowing relevant cells (e.g., vessel endothelia, platelets, WBC); No histological stains 4,5

Cyclic multiplexed-traditional stains, destain

/restain Mapping or co-detection of tissue structures or cells reflected in different stains on a single section Routinely prepared histological sections Conventional ways of the stain, destaining, restaining Upright microscopes; imaging process software Low cost, easy to handle. Compatible with standard equipment and procedures.

Multiplexed stains Low multiplex; Difficulty with image blending; Low throughput; Manual operation; Time-consuming; Risk of damaged section; Manual operation 6,7,8

Artificial intelligence (AI) based Digital imaging Tissue structures and cellar components are stained in the style label-free on a single section. Routinely prepared histological sections for deep learning of AI Virtual HE or histological special stains Confocal or conventional microscopes in a variety, AI software Multiplexed stains in a single section; No staining toxic; quick procedure; low cost for high throughput; Automation High technique threshold; Limited to traditional histological stains 9,10,11,12,13

Spectral multiplexed, single-step Localization of tissue structures or cells with IF for various targets on a single section Routinely prepared histological sections; labeling antibodies and validation IF Advanced /confocal microscope; Special software Single-step, Straightforward; Compatible with standard equipment and procedures

 More time consuming; advanced apparatus; Limited to IF; Relatively low resolution 14,15

Co-detection (CODEX) Mapping cellar heterogeneity at the single-cell level with multiplexed IF on a single section Routinely prepared histological sections; antibodies conjugated with oligonucleotide barcodes and validation IF with the complementary oligonucleotide barcodes labeled with fluorophores Fluorescent microscope; Special software High resolution; High multiplex; Low cross-reactivity; Spatial information of targets; Quick method; Automation Relative expensive; Presence of autofluorescence; Validation of labeling antibodies; Limited to IF; Difficult in connection or blending with traditional stains 16,17,18,19,20 

Digital spatial profiling (DSP) Spatial information on targeted protein and RNA in a single section Routinely prepared histological sections; antibodies conjugated with oligonucleotide barcodes and validation Traditional ways of immune stains; complementary oligonucleotide barcodes Computer, Nano-string nCounter, Image analysis-related wares High resolution; High multiplex; high throughput; Absence of autofluorescence; Automation Lack of single-cell resolution and spatial information; Limited to the region of interest; Absence of an image; Time-consuming 16,21,22

InsituPlex, single step Mapping tissue structures or cellar phenotypes on a single section Routinely prepared histological sections; antibodies conjugated with oligonucleotide barcodes and validation IF with the complementary oligonucleotide barcodes labeled with fluorophores Fluorescent microscope Single step; Compatible with standard workflows; Automation Time-consuming; Limited to IF. (Limited data available) 23,24

Chromgen labeled Cyclic multiplexed-imaging Localization of tissue structures or cellar phenotypes on a single section. Routinely prepared histological sections Conventional ways of IHC; Removal of signals, restain Upright microscopes, and imaging processing software Low cost; Straightforward forward; Easy to handle; Compatible with standard equipment and procedures Time-consuming; Limited to IHC; Low multiplex; Difficulty with image blending; Low throughput; Manual operation; Risk of the damaged section 25,26

IMC Mapping tissue structures and cellar heterogeneity on a single section at a single-cell level Routinely prepared histological sections; antibodies conjugated with isotope metals Immune staining; laser ablation Mass spectrometer; Hyperion Tissue Imager; Special software Automation; Low background noise; Quantification Slow imaging; Destructive detection for the tissue; Expensive; Limited to ROI; Low sensitivity; Risk of cross-contamination 14,27, 28

MIBI Tissue imaging on a single section at a single-cell level Routinely prepared histological sections; antibodies conjugated with isotope metals Staining/oxygen primary-ion ablation Mass spectrometer; Hyperion Tissue Imager; Special software Automation; Low background noise; High resolution; High multiplex; Quantification Slow imaging; Destructive detection for the tissue; expensive; Limited to ROI; low sensitivity; Risk of cross-contamination 27,30,31

Microfluidics- IF Rapid detection of contents in liquid A discrete droplet of liquid to be detected Directly labeling Well-plate reader; Fluorescent microscope; spectrometer Software Simple; easy-to-operate; Inexpensive Special chip-integrated device; Limited to liquid contents detection 32,33

References

1. Piccinin MA, Schwartz J. Histology, Verhoeff Stain. 2022 May 8. In: StatPearls [Internet]. Treasure Island (FL): StatPearls Publishing; 2022 Jan–. PMID: 30085592.

2. Gurina TS, Simms L. Histology, Staining. 2022 May 8. In: StatPearls [Internet]. Treasure Island (FL): StatPearls Publishing; 2022 Jan–. PMID: 32491595.

3. Alturkistani HA, Tashkandi FM, Mohammedsaleh ZM. Histological Stains: A Literature Review and Case Study. Glob J Health Sci. 2015 Jun 25;8(3):72-9. doi: 10.5539/gjhs.v8n3p72. PMID: 26493433; PMCID: PMC4804027.

4. Fallon ME, Mathews R, Hinds MT. In Vitro Flow Chamber Design for the Study of Endothelial Cell (Patho)Physiology. J Biomech Eng. 2022 Feb 1;144(2):020801. doi: 10.1115/1.4051765. PMID: 34254640; PMCID: PMC8628846.

5. Zhang C, Neelamegham S. Application of microfluidic devices in studies of thrombosis and hemostasis. Platelets. 2017 Jul;28(5):434-440. doi: 10.1080/09537104.2017.1319047. Epub 2017 Jun 5. PMID: 28580870; PMCID: PMC5819608.

6. Hinton JP, et al. A Method to Reuse Archived H&E Stained Histology Slides for a Multiplex Protein Biomarker Analysis. Methods Protoc. 2019;2(4):86. doi: 10.3390/mps2040086. 

7. Bolognesi MM, Manzoni M, Scalia CR, Zannella S, Bosisio FM, Faretta M, Cattoretti G. Multiplex Staining by Sequential Immunostaining and Antibody Removal on Routine Tissue Sections. J Histochem Cytochem. 2017;65(8):431-444. doi: 10.1369/0022155417719419.

8. Ozawa A, Sakaue M. New decolorization method produces more information from tissue sections stained with hematoxylin and eosin stain and masson-trichrome stain. Ann Anat. 2020;227:151431. doi: 10.1016/j.aanat.2019.151431. 

9. Rivenson Y, Wang H, Wei Z, de Haan K, Zhang Y, Wu Y, et al. Virtual histological staining of unlabelled tissue-autofluorescence images via deep learning. Nat Biomed Eng. 2019;3(6):466-477. doi: 10.1038/s41551-019-0362-y. 

10. Zhang Y, de Haan K, Rivenson Y, Li J, Delis A, Ozcan A. Digital synthesis of histological stains using micro-structured and multiplexed virtual staining of label-free tissue. Light Sci Appl. 2020;9:78. doi: 10.1038/s41377-020-0315-y

11. Chen Z, Yu W, Wong IHM, Wong TTW. Deep-learning-assisted microscopy with ultraviolet surface excitation for rapid slide-free histological imaging. Biomed Opt Express. 2021;12(9):5920-5938. doi: 10.1364/BOE.433597. 

12. de Haan K, Zhang Y, Zuckerman JE, Liu T, Sisk AE, Diaz MFP, et al. Deep learning-based transformation of H&E stained tissues into special stains. Nat Commun. 2021;12(1):4884. doi: 10.1038/s41467-021-25221-2

 13. Kavitha MS, Gangadaran P, Jackson A, Venmathi Maran BA, Kurita T, Ahn BC. Deep Neural Network Models for Colon Cancer Screening. Cancers (Basel). 2022 Jul 29;14(15):3707. doi: 10.3390/cancers14153707. PMID: 35954370; PMCID: PMC9367621.

14. EPK, et al. The RareCyte® Platformfor Next-Generation Analysis of Circulating Tumor Cells. Cytometry A (2018) 93:1220–5. doi: 10.1002/cyto.a.23619

15. Levin M, Flor AC, Snyder H, Kron SJ, Schwartz D. UltraPlex Hapten-Based Multiplexed Fluorescent Immunohistochemistry. Methods Mol Biol (2021) 2350:267–87. doi: 10.1007/978-1-0716-1593-5_17

16. De Smet F, Antoranz Martinez A, Bosisio FM. Next-Generation Pathology by Multiplexed Immunohistochemistry. Trends Biochem Sci (2021) 46:80–2. doi: 10.1016/j.tibs.2020.09.009

17. Tan WCC, Nerurkar SN, Cai HY, Ng HHM, Wu D, Wee YTF, Lim JCT, Yeong J, Lim TKH. Overview of multiplex immunohistochemistry/immunofluorescence techniques in the era of cancer immunotherapy. Cancer Commun (Lond). 2020 Apr;40(4):135-153. doi: 10.1002/cac2.12023. Epub 2020 Apr 17. PMID: 32301585; PMCID: PMC7170662.

18. Goltsev Y, Samusik N, Kennedy-Darling J, et al. Deep Profiling of Mouse Splenic Architecture with CODEX Multiplexed Imaging. Cell. 2018;174:968–981.e915. 

19. Park J, Liu C, Kim J, Susztak K. Understanding the kidney one cell at a time. Kidney International. 2019;96:862–870.

20. Black S, Phillips D, Hickey JW, Kennedy-Darling J, Venkataraaman VG, Samusik N, et al. CODEX Multiplexed Tissue Imaging With DNAConjugated Antibodies. Nat Protoc (2021) 16:3802–35. doi: 10.1038/ s41596-021-00556-8

21. Hernandez S, Lazcano R, Serrano A, Powell S, Kostousov L, Mehta J, Khan K, Lu W and Solis LM (2022) Challenges and Opportunities for Immunoprofiling Using a Spatial HighPlex Technology: The NanoString GeoMx® Digital Spatial Profiler. Front. Oncol. 12:890410. doi: 10.3389/fonc.2022.890410 

22. Van TM, Blank CU. A user's perspective on GeoMxTM digital spatial profiling. Immunooncol Technol. 2019 May 30;1:11-18. doi: 10.1016/j.iotech.2019.05.001. PMID: 35755324; PMCID: PMC9216425.

23. Manesse, M.; Patel, K.K.; Bobrow, M.; Downing, S.R. The InSituPlex® Staining Method for Multiplexed Immunofluorescence Cell Phenotyping and Spatial Profiling of Tumor FFPE Samples. Methods Mol. Biol. 2019, 2055, 585–592.

24. Mohammed, A.M.; Xia, Z.; Chatterjee, G.; Hwang, K.; Manesse, M. Abstract 1183: High-plex spati al profiling of whole FFPE tissue sections using InSituPlex® technology for discovery applications. Cancer Res. 2019, 79, 1183. 

25. Remark R, Merghoub T, Grabe N, Litjens G, Damotte D, Wolchok JD, et al. In-Depth Tissue Profiling Using Multiplexed Immunohistochemical Consecutive Staining on Single Slide. Sci Immunol (2016) 1(1):aaf6925. doi: 10.1126/sciimmunol.aaf6925 

26. Akturk G, Sweeney R, Remark R, Merad M, Gnjatic S. Multiplexed Immunohistochemical Consecutive Staining on Single Slide (MICSSS): Multiplexed Chromogenic IHC Assay for High-Dimensional Tissue Analysis. Methods Mol Biol (2020) 2055:497–519. doi: 10.1007/978-1- 4939-9773-2_23

27. Bosisio FM, Van Herck Y, Messiaen J, Bolognesi M, Marcelis L, Van Haele M, Cattoretti G, Antoranz A and De Smet F (2022) Next-Generation Pathology Using Multiplexed Immunohistochemistry: Mapping Tissue Architecture at Single-Cell Level. Front. Oncol. 12:918900. doi: 10.3389/fonc.2022.918900

28. Giesen C, Wang HA, Schapiro D, Zivanovic N, Jacobs A, Hattendorf B, et al. Highly Multiplexed Imaging of Tumor Tissues With Subcellular Resolution by Mass Cytometry. Nat Methods (2014) 11:417–22. doi: 10.1038/ nmeth.2869 

29. Baharlou H, Canete NP, Cunningham AL, Harman AN, Patrick E. Mass Cytometry Imaging for the Study of Human Diseases-Applications and Data Analysis Strategies. Front Immunol (2019) 10:2657. doi: 10.3389/ fimmu.2019.02657

30. Keren, L.; Bosse, M.; Thompson, S.; Risom, T.; Vijayaragavan, K.; McCaffrey, E.; Marquez, D.; Ang oshtari, R.; Greenwald, N.F.; Fienberg, H.; et al. MIBI-TOF: A multiplexed imaging platform relates cellular phenotypes and tissue structure. Sci. Adv. 2019, 5, eaax5851.

31. Keren, L.; Bosse, M.; Marquez, D.; Angoshtari, R.; Jain, S.; Varma, S.; Yang, S.-R.; Kurian, A.; Van Valen, D.; West, R.; et al. A Structured Tumor-Immune Microenvironment in Triple Negative Breas t Cancer Revealed by Multiplexed Ion Beam Imaging. Cell 2018, 174, 1373–1387.e19.

32. Zhang Y. Magnetic Digital Microfluidics for Point-of-Care Testing: Where Are We Now? Curr Med Chem. 2021;28(31):6323-6336. doi: 10.2174/0929867327666200903115448. PMID: 32881657.

33. Wang X, Hong XZ, Li YW, Li Y, Wang J, Chen P, Liu BF. Microfluidics-based strategies for molecular diagnostics of infectious diseases. Mil Med Res. 2022 Mar 18;9(1):11. doi: 10.1186/s40779-022-00374-3. PMID: 35300739; PMCID: PMC8930194.

COMMENT 13:

Major comments:

The work seems similar to the existing technologies e.g., Co-detection by indexing (CODEX). The author requires to justify the novelty of the present work with the similar existing technologies emphasizing the novelty of current work. 

RESPONSE:

That is an in-depth issue. 

Recently, co-detection by indexing (CODEX) multiplexed immunofluorescence for over 50 antigens was developed (1,2). The technique employs oligonucleotide barcodes which are conjugated to primary antibodies to improve multiplexed imaging. All the conjugated primary antibodies are incubated and fixed to either a routinely prepared frozen or paraffin-embedded section. Complementary oligonucleotide barcodes connected to fluorophores are serially added, imaged, and removed to generate highly multiplexed imaging datasets without significant tissue degradation. It aims to explore cellular heterogeneity at molecular and cellular levels (e.g., cell mapping) (3,4). Here, we demonstrate the technique – FCS, which complies with the current staining workflow. But it uses a flow chamber to perform multiplexed stains in a single routinely prepared section. These stains include traditionally routine HE and special histological stains, and IF (or IHC). The technique focuses on tissue structure or cellular heterogeneity reflected in various stains at tissue and cellular levels. Although they are both antibody-based multiplexed detection methods to some extent the most challenging work – the validation of antibody labeling for use of CODEX was omitted in FCS. Therefore, we can conclude that 1) in goals: FCS performs multiplexed stains to explore tissue structure or cellular heterogeneity reflected in multiplex stains (e.g., HE, histological special stains, IF, etc), and in the contrast, CODEX aims to detect cellular heterogeneity displayed in multiplexed immunofluorescence; 2) on convenience: the labeling antibodies with oligonucleotide barcodes and subsequent used- Complementary oligonucleotide barcodes must be validated before employed in CODEX, and on the contrary, the commercial antibodies are ready-use for FCS; 3)in consumption, very expensive for CODEX, and low cost for FCS; 4) in compatibility with other equipment or staining: difficult for CODEX and easy for FCS. The novelty of the current work was mentioned briefly as well in Abstract section. For more details, please also refer to Table 1 on the previous page.

Besides thanks to the considerable prompt, we have added some descriptions of the recent advancements related to the current work in Introduction section. The addition is marked with red. Please see the red text on Lines 83 and 84, page 3.

References 

1. Goltsev Y, Samusik N, Kennedy-Darling J, et al. Deep Profiling of Mouse Splenic Architecture 

with CODEX Multiplexed Imaging. Cell. 2018;174:968–981.e915. [PubMed: 30078711] 

2. Park J, Liu C, Kim J, Susztak K. Understanding the kidney one cell at a time. Kidney International. 

2019;96:862–870.

3. Phillips D, Schürch CM, Khodadoust MS, Kim YH, Nolan GP and Jiang S (2021) Highly Multiplexed Phenotyping of Immunoregulatory Proteins in the Tumor Microenvironment by CODEX Tissue Imaging. Front. Immunol. 12:687673. doi: 10.3389/fimmu.2021.687673.

4. Neumann EK, Patterson NH, Rivera ES, Allen JL, Brewer M, deCaestecker MP, Caprioli RM, Fogo AB, Spraggins JM. Highly multiplexed immunofluorescence of the human kidney using co-detection by indexing. Kidney Int. 2022 Jan;101(1):137-143. doi: 10.1016/j.kint.2021.08.033. Epub 2021 Oct 5. PMID: 34619231; PMCID: PMC8741652.

COMMENT 14:

Major comments:

How did the author determine the flow rate to be 1 ml/min, and did they compare this value to other flow rates? If yes, please provide a brief description of the observation in the results section.

RESPONSE:

That is a good point.

In our trial tests, we optimized the flow rate before the formal experiments were performed. The optimization is based on four criteria – tissue integration after perfusion, turnaround time, solution volume used, and the coordination of relevant apparatuses. The flow rates to be tested are 0.5, 1, 1.5, and 2 ml/min. The trial tests were on the sections of mouse heart tissue (5 µm in thickness) by 1 hour- perfusion. 

The preliminary outcomes show that there are no shear impairments of the section imposed by the various speeds of chamber flows mentioned.

It is really a waste of chemicals in FCS, especially for costly antibodies under the perfusion of 1.5 or 2 ml/min. The flow rate of 0.5 ml/min had a longer turnaround time (slow washing, long time for transition from the reservoir to the chamber).

After balancing all the factors, the flow rate of 1 ml/min was selected by virtue of relatively short turnaround time, small solution volume, and the good coordination of relevant apparatuses (e. g., tubings size/length, containers of staining solutions).

We have provided a brief description in the Results Section. The addition is labeled with red (please see Lines 171-175, Page 6).

Reviewers' comments:

Reviewer #1: 

COMMENT 1:

The authors presented an interesting approach for multiplex staining under the title “Flow chamber staining modality for real-time inspection of dynamic phenotypes in multiple histological stains”. The authors tried to address the current limitation of traditional immunohistology by developing a new technique that has multiple advantages compared to the traditional staining methods. Additionally, the authors reported that their technique is very useful in the situation of scarce tissue availability with real-time imaging and recording. 

RESPONSE:

The encouraging comment is greatly appreciated. Thanks!

COMMENT 2:

Major concerns:

1- What is the novelty of this approach compared to the already available commercial products for automated immunohistology, especially the authors mentioned in the discussion “Fortunately, more recently, commercially produced flow chambers (Applied BioPhysics, Bioptechs, C & L Instruments, Inc., Fluxion, Glycotech Corporation, ibidi, Provitro, Stovall Life Science, Inc. Warner Instruments, Inc.) in a variety of geometries are available that potentially overcomes these weaknesses. What about the other factors that usually affect the staining procedure such as temperature and light?

RESPONSE:

That is a good issue. 

For the first query (What is the novelty of this approach…..), we can find out the novelty of FCS in comparison with present commercially-produced flow chambers. The differences or novelties are summarized in the following table.

 Traditional Flow Chamber Technique Flow Chamber staining(FCS)

Purposes Establishing invivo-like conditions for the cultivation of cells under flow or observation of interplay between coated reagents or cells and flow. Histological, immune, or in situ Hybrid (to be tested) stains on a single section.

Applications Endothelial cells relevant to moving flow (e.g., vessels) or other coating substances. All kinds of sections from normal or pathological specimens.

Observation modalities Inverted microscopes. Upright/inverted microscopes.

Findings Morphological changes of living cells, in response to different flow shears, are reflected by imaging or immunofluorescence. Various phenotypes of tissues or cells, reflected by different staining colors or immunofluorescence. 

For the novelty of FCS on the histological staining, we can make a comparison with traditional staining. Those have been mentioned in Abstract Section. Here we brief, 1) quick switching to different stains, 2) a live observation and photographing, and 3) an efficient blend of images. 

For the second query (What about the other factors….), we would say that the factors which affect the staining procedure in conventional staining fashions, will also affect the staining procedures in Flow chamber staining (FCS), such as temperature and light mentioned since the principle behind is same. However, the specials on FCS are the solution flow rate, bubble formation in the chamber, the thickness of coverslips used for section support, and maybe imaging resolution. 

The bubble formation has been discussed in the Discussion Section. 

For the solution flow rate, it seems to us, but not tested, that the high rate of the flow leads to a speedy stain perhaps due to quick diffusion.

For the thickness of coverslips, we used the coverslips with a thickness of 0.01 mm. A thickness over 0.01mm would affect the image resolution at the higher magnification (> or = 40 magnification in the objective lens).

For the resolution, the mounting medium (also known as refractive index matching medium) aims to make tissue transparent by minimizing light scattering and light absorption, thus allowing for a high-resolution image. Without the mounting medium as in FCS, the resolution degradation problem would maybe appear on the higher magnification (> or = 40 magnification in the objective lens) of traditional stains (e.g., HE). However, this problem does not seem to bother us during the work because a 20x objective lens is enough to see the details and neither happens in the observation of Immunofluorescence staining on the higher magnification (> or = 40 magnification in the objective lens). In our view, the reason is that flow solutions that have infiltered the tissue inspected should be regarded as one kind of mounting medium somehow.

COMMENT 3:

Major concerns:

2- One of the important aspects for evaluation of the new modality compared to the traditional methods is to assess the time used to complete the protocol of staining. Can the authors elaborate more on that compared to the traditional staining? Is it significantly shorter, longer or the same?

RESPONSE:

The issue is a completely reasonable query. The answers are not easy. They depend on the throughput of staining sections. 

For a section, the time (inclusive of destaining) used to complete the staining with FCS is much shorter than that taken in the corresponding traditional staining. For example, we took about 85 min (Please refer to Fig. 4-1) to complete 3 kinds of staining, inclusive of HE, PAS, Sirius Red, and the interval destaining processes. The same stains in the traditional way on the adjacent sections would take about 100 min (Please see Table S2) to be completed, exclusive of destaining processes. Thus FCS is highly efficient for one section in comparison with the corresponding traditional stains. 

For high throughput of staining sections, the time used to complete the staining with FCS is relatively longer than that taken in the corresponding traditional staining. The higher throughput of sections to be stained, the less efficient FCS is in comparison with the corresponding traditional stains. 

Therefore, FCS is very highly efficient for scarce samples, with intention of intensive investigation. Application of FCS is not encouraged for high throughput of staining sections.

Thanks to the kind reminder, we have added some descriptions in Discussion Section. Please see the green text on lines 449, Page 14, and 459-461, Page 15.

COMMENT 4:

Major concerns:

3- The authors mentioned “By the technique, staining processing can be videoed and made a backup for off-site pathologists, which facilitates tele-consultation or -education in current digital pathology”, I suggest adding a short video that can summarize the steps of the staining protocol to strengthen the authors claim and for reproducibility, especially that there is only schematic diagram in fig.1 and fig. S1without digital imaging of the setup itself.

RESPONSE:

Yes, that is a good idea. We have prepared additional videos in brief since the space taken by the original video is very large and it is very difficult to upload a larger video (we have tried one time). Please see the additional description in green in Lines 499, page 16, and 535-536, page 17, and videos prot1, prot2, and prot3.

COMMENT 5:

Major concerns:

4- In the abstract, the authors should state the gap and the current limitation in the conventional staining protocols.

RESPONSE:

Yes, we have made amendments in Lines 34-37, Page 2, marked with a green, to address the reviewer’s kind suggestions.

COMMENT 6: 

Major concerns:

5- In the Introduction, it is little redundant, and it can be more concise with supporting literature especially when the authors mentioned the disadvantages of multiplex staining. Additionally, the authors elaborated more on their methodology in the last paragraph which is more relevant to the method section.

RESPONSE:

We agree with the reviewer. Efforts have been made to concise the description in the Introduction Section. Please refer to the green marked text in Lines 98,102,103,116,126-128, Page 4, and 133,134,136, Page 5.

COMMENT 7:

Major concerns:

6- In the results: the authors mentioned “Images generated by IF and HE of FCS and traditional staining procedures in the adjacent sections were demonstrated” Figs S4-S6, I noticed 1- that most of the sections used are not matching, were the sections used for both methods are identical? 2- Additionally, H&E is better resolution or quality in traditional staining. 3- Although the signal localization is close in IF but there are a lot of differences in IF stained sections in both methods, please check CD31 in Fig. S4, this would be problematic in specific small target regions assuming that the staining was done on identical sections for validation.

RESPONSE:

Thank the reviewer for the interesting and in-depth issues. 

For the first query (most of the sections used are not matching, were the sections used for both methods identical?), we would say that the sections used for both methods are not identical, but we focused on the corresponding regions in adjacent sections that have similar structures.

And the matching degree for the two images depends on 1) the homogeneity of structures and cells in the tissue to be inspected, 2) section thickness, 3) the magnification of the objective lens, and 4) imaging resolution. 

1) Homogeneity of structures and cells in the tissue to be inspected. The high similarity of contents of the tissue (e.g., muscle, heart tissue) is apt to be a high degree match between the adjacent sections, and the high heterogeneity of contents (such as liver, kidney, etc.), on the contrary, is likely to be a low degree match between the adjacent sections.

2) Section thickness. Although we prepared the same thick sections (5µm) in adjacence for both methods. The matching degree would be high in thinner thickness, and vice versa.

3) Magnification of objective lens. We observed the outcomes at a magnification of x20 (objective lens) for both under the same microscope. But the matching degree would be low under higher magnification, and vice versa since more details can be examined at a higher magnification.

 4) Imaging resolution. The matching degree would be high in a lower resolution, and vice versa since more details can be examined at a higher resolution. For this issue, we also mentioned above. Please see Response to Comment 2.

For the above-mentioned reasons, we would say that the statistic equal is more important as we described in Figs 3-3,5-2 and 6. Sorry to say so.

For the second query (H&E is better resolution or quality), we agree with the reviewer. Traditional stains of HE display better resolution or quality especially under higher magnification in comparison with FCS. But FCS gives out stronger signals in immunofluorescence. Please see also Response to Comment 2. 

For the third query (……this would be problematic in specific small target regions …..), the reviewer is right. But we have no better ways to see both contour profiles and details in one image for the adjacent sections stained with divergent staining fashions. In order to display more details of the signals, we focused on the specific small target regions with the indicated region of interest on the organ’s contour.

COMMENT 8:

Major concerns:

7- The authors should refine the results section and focus on their results only, some of the results text is more related to discussion.

RESPONSE: 

We consent to the reviewer’s good suggestion. We have updated the results section in the manuscript. Please see the modification in green of Lines 166,167,177, Page 6; 207-213, Page 7; 239,240,245-252, Page 8; and 270,271, Page 9. 

COMMENT 9:

Major concerns:

8- The discussion section has a lot of redundancy similar to the results section. The authors should summarize their key findings and analyze how their results fit in with previous research of similar modalities and what is the added values. Also, some of the discussion is related mostly to suggestion and potential use of the modality which isn’t supported by data in the current study for example “FCS staining can normalize the stain quality with automatic control of the flow speed……….”.

RESPONSE:

We agree with the reviewer’s kind suggestion. We have updated the discussion section in the manuscript. Please see Lines 353,376-380, Page 12; 394-398, Page 13; and 459-461, Page 15. The changes are labeled with green.

COMMENT 10:

Major concerns:

9- The authors already mentioned a lot of limitation of FCS modality which can be limiting factor to adopt this technique. Did the authors test more than one antibody with this technique?

RESPONSE:

That is a good issue. Thanks! 

To test more than one antibody with this technique is theoretically feasible but we did not test. The current work is limited to single-target detection combined with conventional histological stains under a common-used microscope. But it is one of the goals we pursue in the future to broaden FCS application. 

Multiplex imaging can be carried out to target multi-proteins in the same section. It is one of the most important advantages in the field of morphology.

We are just opening a small slot in the window and the perfection of the technique will continue with time on. For example, the automatic and intellectual technique may be realized with flow chamber staining in near future, thus making FCS a lower technique threshold, and the stains by FCS will become simple and easier to be accepted.

COMMENT 11:

Major concerns:

10- On of the most important advantages of the multiplex imaging and other new techniques is to target multi-protein or use multiple antibodies on the same section? Is it possible with the use of FCS?

RESPONSE:

The reviewer is right. Using multiple antibodies and Multiplex imaging can detect multi-proteins in the same section (please refer to the responses to Comments 12/13 of Additional Editor/Reviewer 2). It is important in the days of molecular science. The application of FCS fused with this technique is possible.

We have just completed the initial step of FCS and there are still a lot of things to be done for perfection.

COMMENT 12:

Minor comments:

11- In figure 1 the authors demonstrated schematic diagram for the workflow, which is starting with pump, it’s little confusing with arrows direction. How is the syringe withdrawing from the reservoir, although the reservoir is positioned post or after the chamber?

RESPONSE:

The reviewer’s confusion maybe comes from our unclear figure. So we have improved the quality. Please see the updated Figure 1.

For the syringe pump, we used Harvard Apparatus (Pump 11). It has four functions - Infuse, Withdraw, Infuse/Withdraw and Withdraw/Infuse. The withdraw-only function was employed in the current experiment.

COMMENT 13:

Minor comments:

12- Please add scale bars for all the images, especially supplementary figures.

RESPONSE:

OK. No problem. The scale bars have been added in the appropriate figures. Please refer to S6 and S7 Figs in Supporting Information.

COMMENT 14:

Minor comments:

13- The authors mentioned about supplementary videos, but I couldn’t find.

RESPONSE:

Sorry for that. The videos we uploaded are MP4 files (file.MP4). In general, they are easy to be opened on a domestic PC. Maybe they are a little too large. We clearly remembered that the uploading of the videos took a very long time (about 3 hrs) during the initial submission.

---

## [Decision Letter · Decision Letter 1]

12 Mar 2023

PONE-D-22-32143R1Flow chamber staining modality for real-time inspection of dynamic phenotypes in multiple histological stainsPLOS ONE

Dear Dr. Li,

Thank you for submitting your manuscript to PLOS ONE. After careful consideration, we feel that it has merit but does not fully meet PLOS ONE’s publication criteria as it currently stands. Therefore, we invite you to submit a revised version of the manuscript that addresses the points raised during the review process.

We look forward to receiving your revised manuscript.

Kind regards,

Kanhaiya Singh, Ph.D

Academic Editor

PLOS ONE

Journal Requirements:

Additional Editor Comments:

Please address the remaining concerns of Reviewer 1.

Reviewers' comments:

Reviewer's Responses to Questions

**Comments to the Author**

1. If the authors have adequately addressed your comments raised in a previous round of review and you feel that this manuscript is now acceptable for publication, you may indicate that here to bypass the “Comments to the Author” section, enter your conflict of interest statement in the “Confidential to Editor” section, and submit your "Accept" recommendation.

Reviewer #1: (No Response)

Reviewer #2: All comments have been addressed

2. Is the manuscript technically sound, and do the data support the conclusions?

Reviewer #1: Yes

Reviewer #2: Yes

3. Has the statistical analysis been performed appropriately and rigorously? 

Reviewer #1: Yes

Reviewer #2: Yes

4. Have the authors made all data underlying the findings in their manuscript fully available?

Reviewer #1: Yes

Reviewer #2: Yes

5. Is the manuscript presented in an intelligible fashion and written in standard English?

Reviewer #1: Yes

Reviewer #2: Yes

6. Review Comments to the Author

Reviewer #1: I would like to thank the authors for their detailed responses, mostly they addressed all the comments, but I still have some minor comments:

Major concerns:

1- For my 1st query, I think the authors should add the information mentioned in response to reviewers to the manuscript but in brief, especially if there is any variables or factors that would affect the reproducibility or the quality of the results.

2- Again, I couldn’t find any videos in submitted revised manuscript.

3- In the results: the authors mentioned “Images generated by IF and HE of FCS and traditional staining procedures in the adjacent sections were demonstrated” Figs S4-S6, I noticed. 3- Although the signal localization is close in IF but there are a lot of differences in IF stained sections in both methods, please check CD31 in Fig. S4, this would be problematic in specific small target regions assuming that the staining was done on identical sections for validation. Please add your response to the discussion section specifically the 3rd query, may be additional to the limitation.

4- The authors already mentioned a lot of limitation of FCS modality which can be limiting factor to adopt this technique. Did the authors test more than one antibody with this technique? Please indicate that the current work is limited to single-target detection combined with conventional histological stains under a common-used microscope in the manuscript.

Reviewer #2: I am satisfied with the response of authors. The authors clearly demonstrated a novel technique which is a fusion method of traditional histological stains, including immune stains, and conventional flow chamber.

7. PLOS authors have the option to publish the peer review history of their article (what does this mean?). If published, this will include your full peer review and any attached files.

Reviewer #1: No

Reviewer #2: **Yes: **Ajay Kumar Sahi

---

## [Author Response · Author response to Decision Letter 1]

16 Mar 2023

Response to Reviewers

Reviewer #1

COMMENT 1:

I would like to thank the authors for their detailed responses, mostly they addressed all the comments, but I still have some minor comments:

RESPONSE:

We would also like to express our sincere gratitude to the anonymous reviewer for his/her kind patience in helping us to improve the quality of the manuscript. With the insightful comments, the manuscript has significantly been made better. Thanks again.

COMMENT 2:

Major concerns: 1- For my 1st query, I think the authors should add the information mentioned in response to reviewers to the manuscript but in brief, especially if there is any variables or factors that would affect the reproducibility or the quality of the results.

RESPONSE:

That’s a good suggestion. We have added information on the factors that affect the staining outcomes to the revised manuscript. Please see Lines 385-393, Pages 12-13. The addition is marked with red.

COMMENT 3:

Major concerns: 2- Again, I couldn’t find any videos in submitted revised manuscript.

RESPONSE:

That is too bad. We have indeed uploaded the videos and checked the uploading list before approving the submission. Besides, the loading materials have been subjected to technical checks, ensuring everything including files, pictures, and videos is loaded completely and correctly, before handing over to the reviewer’s hands. If something was missing, the relevant staff of Plos one editorial would inform us.

We have mentioned what happened in our cover letter to the editor since we never meet such a case before and can do nothing about it. Maybe it is better for the reviewer her/his self to ask relevant persons of Plos One and they can help.

We are really sorry for that.

 COMMENT 4:

Major concerns: 3- In the results: the authors mentioned “Images generated by IF and HE of FCS and traditional staining procedures in the adjacent sections were demonstrated” Figs S4-S6, I noticed. 3- Although the signal localization is close in IF but there are a lot of differences in IF stained sections in both methods, please check CD31 in Fig. S4, this would be problematic in specific small target regions assuming that the staining was done on identical sections for validation. Please add your response to the discussion section specifically the 3rd query, may be additional to the limitation.

We agree with the reviewer.

We have added the responses to Limitation section. Please see Lines 429, 430, 442-448, Page 14. The addition is marked with red.

Limitation portion had taken a rather large space within the Discussion section after adding if not separated, and therefore, we separated Limitation part from the Discussion section. To keep the manuscript coherent and complete, the isolated becomes a separate section – “5. Limitation” in red (Line 427, Page 14).

COMMENT 5:

Major concerns: 4- The authors already mentioned a lot of limitation of FCS modality which can be limiting factor to adopt this technique. Did the authors test more than one antibody with this technique? Please indicate that the current work is limited to single-target detection combined with conventional histological stains under a common-used microscope in the manuscript.

We agree on the point. We have included the description in the revised manuscript. Please see Lines 360-365, Page12. It is labeled red.

Reviewer #2

COMMENT 1:

I am satisfied with the response of authors. The authors clearly demonstrated a novel technique which is a fusion method of traditional histological stains, including immune stains, and conventional flow chamber.

RESPONSE:

Thank you, Dr. Sahi.

---

## [Decision Letter · Decision Letter 2]

30 Mar 2023

Flow chamber staining modality for real-time inspection of dynamic phenotypes in multiple histological stains

PONE-D-22-32143R2

Dear Dr. Li,

We’re pleased to inform you that your manuscript has been judged scientifically suitable for publication and will be formally accepted for publication once it meets all outstanding technical requirements.

Kind regards,

Kanhaiya Singh, Ph.D

Academic Editor

PLOS ONE

Additional Editor Comments (optional):

Reviewers' comments:

Reviewer's Responses to Questions

**Comments to the Author**

1. If the authors have adequately addressed your comments raised in a previous round of review and you feel that this manuscript is now acceptable for publication, you may indicate that here to bypass the “Comments to the Author” section, enter your conflict of interest statement in the “Confidential to Editor” section, and submit your "Accept" recommendation.

Reviewer #1: All comments have been addressed

2. Is the manuscript technically sound, and do the data support the conclusions?

Reviewer #1: Yes

3. Has the statistical analysis been performed appropriately and rigorously? 

Reviewer #1: Yes

4. Have the authors made all data underlying the findings in their manuscript fully available?

Reviewer #1: Yes

5. Is the manuscript presented in an intelligible fashion and written in standard English?

Reviewer #1: Yes

6. Review Comments to the Author

Reviewer #1: (No Response)

7. PLOS authors have the option to publish the peer review history of their article (what does this mean?). If published, this will include your full peer review and any attached files.

Reviewer #1: No

---

## [Editor Report · Acceptance letter]

11 Apr 2023

PONE-D-22-32143R2 

Flow chamber staining modality for real-time inspection of dynamic phenotypes in multiple histological stains 

Dear Dr. Li:

I'm pleased to inform you that your manuscript has been deemed suitable for publication in PLOS ONE. Congratulations! Your manuscript is now with our production department. 

Kind regards, 

on behalf of

Dr. Kanhaiya Singh 

Academic Editor

PLOS ONE